# What's the Move?
# Hybrid Imitation Learning via Salient Points

**Priya Sundaresan**[*1]**, Hengyuan Hu**[*1]**, Quan Vuong**[2]**, Jeannette Bohg**[1]**, Dorsa Sadigh**[1]
[*]Equal contribution.
[1]Stanford University, [2]Physical Intelligence

## Abstract

While imitation learning (IL) offers a promising framework for teaching robots various behaviors, learning complex tasks remains challenging. Existing IL policies struggle to generalize effectively across visual and spatial variations even for simple tasks. In this work, we introduce **Sphinx** (Salient Point-Based Hybrid ImitatioN and eXecution), a flexible IL policy that leverages multimodal observations (point clouds and wrist images), along with a hybrid action space of low-frequency, sparse waypoints and high-frequency, dense end effector movements. Given 3D point cloud observations, **Sphinx** learns to infer task-relevant points within a point cloud, or *salient points*, which support spatial generalization by focusing on semantically meaningful features. These salient points serve as anchor points to predict waypoints for long-range movement, such as reaching target poses in free-space. Once near a salient point, **Sphinx** learns to switch to predicting dense end-effector movements given close-up wrist images for precise phases of a task. By exploiting the strengths of different input modalities and action representations for different manipulation phases, **Sphinx** tackles complex tasks in a sample-efficient, generalizable manner. Our method achieves **86.7**% success across 4 real-world and 2 simulated tasks, outperforming the next best state-of-the-art IL baseline by **41.1**% on average across **440** real world trials. **Sphinx** additionally generalizes to novel viewpoints, visual distractors, spatial arrangements, and execution speeds with a **1.7**× speedup over the most competitive baseline. Our website contains code for data collection and training code along with supplementary videos: http://sphinx-manip.github.io.

## 1 Introduction

Imitation learning (IL) of visuomotor policies is a widely used framework for teaching robots manipulation tasks given demonstrations collected by humans (Schaal, 1996). While prior works have shown that IL policies can learn a range of behaviors with sufficient data, from simple object pick-and-place to more complex tasks, they typically succeed only in highly controlled settings with low variation. Generalizing to realistic visual and spatial variations remains a significant challenge. Consider teaching a robot to make a cup of coffee in the morning, which demands precision, long-horizon reasoning, and tolerance to environment variations. The robot must first carefully grasp a mug handle, position it under the machine, insert a pod into a narrow slot, close the lid, and press a button – all with very little margin for error (Fig. 1). Even after mastering this sequence, the policy might struggle with *spatial* changes like moving the machine, or *visual* changes such as new coffee pods, spilled grounds, a different camera angle, or varying lighting conditions. This underscores the need for IL policies that can learn complex tasks from a limited number of demonstrations while effectively generalizing to natural and expected variations in the real-world.

Conventional IL policies often struggle with both performance and generalization, largely due to limitations in their input and output representations. First, they tend to rely heavily on visual inputs like RGB images, treating irrelevant details like the background, lighting, or viewpoint the same as task-relevant information. This can cause a policy to memorize specific scenes, making it brittle to *visual* variations (Zhao et al., 2023; Chi et al., 2023). Second, these policies usually predict actions

---

Correspondence to { priyasun@stanford.edu, hengyuan.hhu@gmail.com }

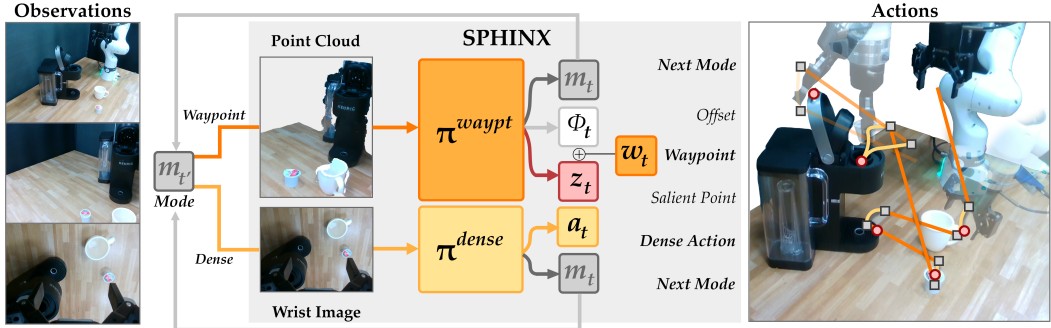

Figure 1: **SPHINX** is a hybrid IL agent which learns to switch amongst different modes ($m_t$) of execution to tackle complex tasks with visuospatial generalization. In *waypoint* mode, $\pi^{\text{waypt}}$ takes a point cloud as input, and predicts a single *waypoint* $w_t$ as an offset ($\phi_t$) to a task-relevant *salient point* $z_t$ (i.e. mug handle, coffee pod, etc. denoted). After reaching a waypoint via a controller, the policy uses learned switching to a dense policy $\pi^{\text{dense}}$, which takes wrist-camera images as input and outputs *dense* actions ($a_t$) for precise manipulation around a salient point. On the right, the policy interleaves both modes of execution to complete a long-horizon coffee-making task guided by salient points (●) and mode switches (■).

for the next immediate timestep, which hampers *spatial* reasoning. Simple spatial movements, like reaching, are predicted through hundreds of end-effector actions, increasing the risk of veering off course. To address these limitations, recent works explore 3D scene representations, such as point clouds and voxel grids, to offer better spatial awareness, and propose predicting actions as end-effector poses (*waypoints*) reachable through a controller or motion planner (Goyal et al., 2023; Sundaresan et al., 2023; Yang et al., 2024a;b; Shridhar et al., 2023). This can drastically shorten the action prediction horizon and enable better spatial generalization. However, these methods often lack precision, as point clouds typically lack the necessary resolution to capture small object details preserved in images. On the other hand, recent image-based IL policies attempt to remedy spatial generalization using *hybrid action spaces* (Belkhale et al., 2023; Shi et al., 2023). Here, the policy has two potential *modes* of execution: *waypoints* for long-range motions like reaching, or *dense* actions — end-effector movements predicted per-timestep — only when precision or reactivity is required. These works ultimately consider training policies with a single input modality type, and optionally hybrid actions. In reality, different phases of a task may lend themselves more favorably to different visual inputs or action modes. A policy which can effectively choose and interchange both the input modality and the underlying action representation during execution remains underexplored.

Our key insight is that by encouraging the policy to attend to *salient points*—task-relevant 3D points—we enable it to choose between different input modalities as well as different action modes when appropriate, improving performance and generalization. Fig. 1 illustrates example salient points in the coffee-making task in red (i.e. the mug handle, pod, etc.). We introduce **SPHINX**: *Salient Point-based Hybrid ImitatioN and eXecution*, a hybrid IL agent which learns to switch amongst a *waypoint policy* which predicts waypoint actions given point clouds, and a *dense policy* which predicts dense actions given close-up wrist-camera images. Specifically, the waypoint policy manages long-range movements by first predicting salient points that narrow the search space of actions around spatially relevant features, promoting *spatial* generalization. It then predicts waypoint actions relative to these points. After reaching a waypoint, **SPHINX** switches to a dense policy which takes wrist-camera images as input. This policy captures close-up object details for precise manipulation and supports *visual* generalization by staying agnostic to broader scene changes. To support training **SPHINX**, we develop a flexible data collection interface that allows demonstrators to specify salient points and switch modes in real-time during teleoperation.

Empirically, we show that **SPHINX** can tackle a range of precise, long-horizon manipulation tasks, including four real-world scenarios (drawer-opening, cup-stacking, coffee-making, toy train assembly) and two simulated ones. **SPHINX** achieves **86.7%** success and outperforms the next best IL baseline by **41.1%** on average, while generalizing better to visual distractors, viewpoints, spatial arrangements, and execution speeds. We open-source our web-based data collection interface for specifying salient points and hybrid teleoperation, alongside code, supplementary material, and videos at: http://sphinx-manip.github.io.

## 2 RELATED WORK

**Imitation Learning for Robotics Control:** Imitation learning has long been a foundational approach in robotics for teaching robots to replicate human demonstrations (Schaal, 1996; Atkeson & Schaal, 1997; Pomerleau, 1988). Robotic imitation learning policies typically take images as input and output motor commands, such as joint positions, velocities, or Cartesian end-effector poses. Recent works of that type (Reuss et al., 2023; Chi et al., 2023; Zhao et al., 2023) have demonstrated strong performance on tasks in controlled settings with a limited initial state distribution. However, they struggle to generalize to unseen visual or spatial variations. To address visual generalization, some works augment vision-based policies with diffusion-generated image observations (Yu et al., 2023; Bharadhwaj et al., 2024). While useful and complementary to our approach, these augmentations do not directly enable spatial generalization. Other works propose replacing image inputs with 3D scene representations such as point clouds and voxel grids, and outputting actions as *waypoints*, 6-DoF poses reachable through motion planning (Sundaresan et al., 2023; Shridhar et al., 2023; Yang et al., 2024b). While this reduces the complexity of action prediction from hundreds of actions to a single pose, 3D representations such as point clouds often lack the resolution to enable precise manipulation of small objects. Other recent approaches like HYDRA (Belkhale et al., 2023) and AWE (Shi et al., 2023) take image inputs but propose a hybrid output action space of *waypoints* and *dense* actions. These distinct action modes are intended for long-range and precise movements, respectively. Our method builds on these approaches by leveraging *salient points* to bridge a hybrid input space of point clouds and wrist-camera images, and a hybrid output action space of waypoints and dense actions.

**Action Representations:** Most visual imitation learning works rely on standard 6-DoF action spaces, but recent efforts explore alternatives for better spatial generalization. One approach involves predicting actions as parameterized manipulation primitives instead of low-level end-effector movements. This reduces the dimensionality of the action space and improves sample efficiency, but often requires task-specific engineering (Dalal et al., 2021; Sundaresan et al., 2023; Nasiriany et al., 2022; Agia et al., 2022). Other methods exploit equivariance, ensuring that transformations of visual inputs (e.g., rotations or scaling) are reflected in output actions (Wang et al., 2024; Yang et al., 2024a;b). However, these works often rely on limiting assumptions like access to object states via segmentation, or single-object tasks. In the grasping domain, many policies consider point clouds as inputs and an output action space defined as per-point predictions for the end-effector pose. This has proven effective for learning sample-efficient and generalizable grasping policies (Saxena et al., 2008; Sundermeyer et al., 2021). Inspired by this, our method also parameterizes waypoint actions as offsets to salient points in a point cloud, but we critically learn a *hybrid* policy which predicts *both* waypoint and dense actions to tackle longer-horizon and precise tasks beyond grasping.

**Data Collection for Imitation Learning:** Despite the advancements in action representations and spatial generalization, the success of visual imitation learning policies still hinges on the quality of teleoperated demonstrations. Human operators typically collect robot data using interfaces like virtual reality controllers (Jedrzej Orbik, 2021), handheld devices (Chi et al., 2024), puppeteering setups (Zhao et al., 2023), or 3D mice (e.g., Spacemouse). However, these interfaces map demonstrator controls directly to robot actions on a *per-timestep* basis, which presents two key limitations. First, the recorded data only captures (observation, dense action) pairs, lacking compatibility with waypoint actions or useful metadata such as salient points. Second, directly controlling long-range movements can be inefficient, noisy, and tiring for demonstrators. To address these issues, we design an interface (Fig. 2) that seamlessly integrates both waypoint and dense action modes. A custom web-based GUI supports waypoint mode, allowing demonstrators to specify salient points and waypoints with the ease of simple clicks and drags. A controller can then reach the specified waypoint automatically, removing the need for constant teleoperation from the demonstrator. Additionally, the interface is compatible with any external device for dense actions, allowing for easy switching between the computer mouse and the device, as long as it is on hand. This provides a flexible and efficient data collection process for high-quality hybrid datasets, with no post-hoc labeling required.

## 3 PROBLEM STATEMENT

In standard IL, we are given a dataset $\mathcal{D}$ of $N$ trajectories of expert demonstrations $\{\tau_1, \ldots, \tau_N\}$. Each trajectory is a sequence of observation action pairs $(o_0, a_0, \ldots, o_T, a_T)$. The goal is to learn

a policy $\pi(a_t|o_t)$ that matches the expert distribution using the following loss $-\mathbb{E}_{\tau \sim \mathcal{D}} \log \pi(a_t|o_t)$. However, this formulation can easily lead to compounding errors for long-horizon tasks where episodes may span hundreds of steps. In this paper, we instead consider a *hybrid* imitation learning setting where the policy can either output a **dense action** $a_t \in \mathbb{R}^7$, a short-range end-effector pose reachable by $t+1$, or a **waypoint** $w_t \in \mathbb{R}^7$, a long-range end-effector pose reachable by a series of interpolated movements. Both waypoint and dense actions capture the end-effector pose, but a waypoint action $w_{t'}$ specified at timestep $t'$ is translated to a sequence of $k_{t'}$ interpolated actions $\{a_{t'}, \ldots, a_{t'+k_{t'}-1}\}$ by a controller $a_t = \mathcal{C}(o_t^{\text{pose}}, w_{t'})$ based on the current pose $o_t^{\text{pose}}$ and the target waypoint $w_{t'}$ specified at $t'$. In practice, we use a simple controller that linearly interpolates between current pose and target waypoint. Then, we can record a timestep in the dataset as $(o_t, a_t, m_t, [w_{t'}])$ where $m_t \in \{\texttt{waypt}, \texttt{dense}, \texttt{terminate}\}$ is the mode and $w_{t'}$ is the optional target waypoint only when $m_t = \texttt{waypt}$. Each $w_{t'}$ spans the next $k_{t'}$ steps decided by the controller. Our goal is thus to learn a hybrid policy $\pi(o_t)$ that first predicts a mode $p(m_t|o_t)$ and then predicts either a waypoint from $\pi^{\texttt{waypt}}(w_t|o_t)$ or a dense action from $\pi^{\texttt{dense}}(a_t|o_t)$.

# 4 SPHINX: SALIENT POINT-BASED HYBRID IMITATION AND EXECUTION

We introduce **SPHINX**: Salient Point-based Hybrid ImitatioN and eXecution, a framework for learning sample-efficient, generalizable imitation policies capable of handling complex, long-horizon manipulation tasks across diverse initial conditions. **SPHINX** combines a high-level waypoint policy $\pi^{\texttt{waypt}}$ for long-range movements and a dense policy $\pi^{\texttt{dense}}$ for precise manipulation (Fig. 1). The waypoint policy takes point clouds as input, classifies semantically meaningful **salient points**, $z_t \in \mathbb{R}^3$, and regresses **waypoint actions** $w_t \in \mathbb{R}^7$. Importantly, $\pi^{\texttt{waypt}}$ predicts the positional component of $w_t$ as an *offset* to the salient point $z_t$. This grounds the desired interaction around a salient point, such as learning to reach for a mug by first identifying the handle. The dense policy $\pi^{\texttt{dense}}$ takes over only for precise actions around a salient point, like carefully inserting a coffee pod into its slot (Fig. 1). Since the waypoint policy handles long-range movements, it uses point clouds to provide spatial context. The dense policy uses wrist camera images as input, capturing detailed object features for precise manipulation and enabling visual generalization to variations in the surrounding scene. Both policies also predict the next mode $m_{t+1}$ to decide which policy to use after completing the current movement. Without loss of generality, we initialize $m_0$ to waypoint mode.

To train **SPHINX**, we first need to collect demonstrations using the two modes and annotate salient point for each waypoint. In Section 4.1, we introduce an intuitive web GUI to easily collect such demonstrations in the hybrid format and record salient points with no additional overhead. Then, we discuss how to learn $\pi^{\texttt{waypt}}(w_t|o_t)$ and $\pi^{\texttt{dense}}(a_t|o_t)$ in Section 4.2 and Section 4.3 respectively.

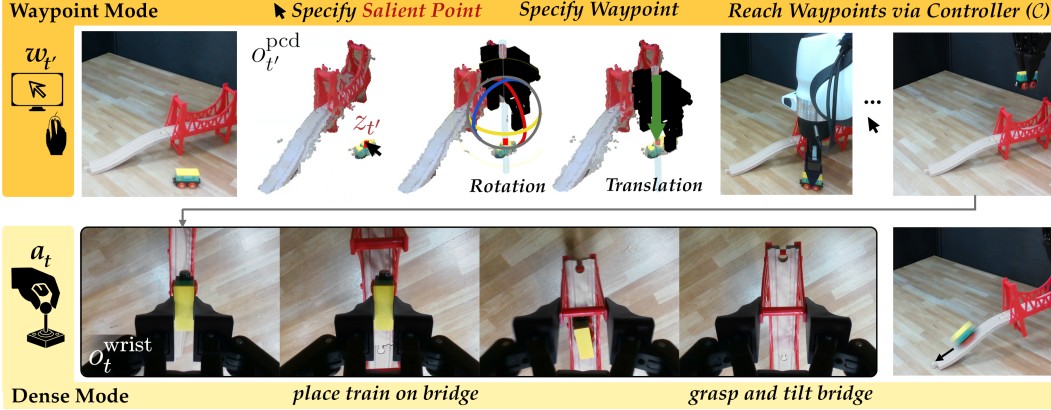

Figure 2: **Data Collection Interface**: The demonstrator visualizes a point cloud $o_{t'}^{\text{pcd}}$ in a web GUI, where they can click a salient point $z_t'$ and specify a waypoint action $w_t'$ by clicking and dragging to rotate or translate a digital twin of the gripper. After the controller $\mathcal{C}$ reaches the waypoint to grasp the train, the process repeats for a waypoint above the bridge. The demonstrator then switches to providing dense actions $a_t$ with a 3D SpaceMouse to carefully place the train on the bridge and tilt it, causing the train to roll.

### 4.1 DATA COLLECTION INTERFACE FOR SPHINX

Without an existing interface that satisfies our need, we design a data collection system to support waypoint specification, salient point annotation, and mode switching seamlessly. Our hardware setup includes two third-person cameras to provide RGB-D observations to construct a colorized point cloud $o_t^{\text{pcd}}$, and one wrist-mounted camera to provide RGB wrist images $o_t^{\text{wrist}}$. We develop a custom web-based GUI for specifying waypoints and salient points in waypoint mode. To provide dense actions instead, a demonstrator can seamlessly switch from the computer mouse to any dense teleoperation device like a VR/game console controller or a 3D mouse (Spacemouse) as in this work.

The top row of Fig. 2 visualizes the web-based GUI and the process of recording a waypoint action. The GUI streams the point cloud of $N$ points $o_t^{\text{pcd}} = \{c_1, c_2, \ldots, c_N\}$ to the browser in *real time* and allows a demonstrator to select a salient point $z_t \in \{1, \ldots, N\}$ for each waypoint by clicking within the point cloud, (e.g. the red dot on the toy car next to the mouse cursor.) After clicking on the salient point, a digital twin of the gripper appears near the salient point to facilitate waypoint specification. The demonstrator can use click and drag interactions on the virtual gripper to set waypoints relative to these salient points. The salient point $\in \mathbb{R}^3$ specifies the *region* of interest for interaction while the waypoint, a 7 DoF target end-effector pose, captures *how* to interact with it. After specifying a waypoint, the linear controller $\mathcal{C}$ defined above interpolates and executes actions to reach the waypoint. Critically, this removes the need for the demonstrator to manually teleoperate long-range movements. The entire waypoint motion is recorded as a sequence $\{(o_t, a_t, \texttt{waypt}, w_{t'}, z_{t'})\}_{t \in \{t', \ldots, t'+k_{t'}\}}$ where $t'$ is the timestep when the waypoint is specified and $k_{t'}$ is the number of steps that the controller takes to complete the waypoint $w_{t'}$.

Once the controller reaches a waypoint, the demonstrator may specify another waypoint or switch to dense mode for precise manipulation. To take over with dense mode, the demonstrator simply operates the teleoperation device, such as a 6DoF joystick, and its movements are automatically detected and mapped to delta end-effector movements. This is illustrated by the bottom row of Fig. 2, where the operator uses the teleoperation device (Spacemouse in this case) to precisely align the toy car above the narrow bridge. Each step in dense mode is recorded as $(o_t, a_t, \texttt{dense})$. Note that regardless of the mode, we record the full set of observations $o_t$ which includes all camera views and proprioception to facilitate data augmentation and compatibility with any IL policy.

### 4.2 THE WAYPOINT POLICY OF SPHINX

The waypoint policy $\pi^{\text{waypt}}$ in SPHINX takes a point cloud $o_t^{\text{pcd}}$ as input and outputs a 7-DoF end-effector pose $w_t$ for the robot to reach via a controller. We utilize point clouds as input to cast part of the action prediction problem to learning a salient map over the points. This encourages the policy to attend to the important spatial features (i.e. the handle on a mug) rather than memorize actions.

The detailed design of the waypoint architecture is illustrated in Fig. 3. At a high level, we would like to have per-point predictions, such as the probability of a point to be salient and the translational offset between the point and the target location of the end-effector, as well as other predictions whose targets are not expressed relative to the points, such as rotation, gripper state, and mode. We use a transformer to process the points and add additional tokens for point-agnostic predictions. We first use farthest-point-sampling (FPS) (Qi et al., 2017) to downsamples a raw point cloud to $D = 1024$ points, and then convert the points $c_i \in \mathbb{R}^6$ to tokens $e_i \in \mathbb{R}^d$ via a shared linear projection layer. Then we feed the entire set of tokens into a transformer (Vaswani et al., 2017; Radford et al., 2019) to get output embeddings. Since the points in a point cloud are unordered, the transformer has no positional embedding and does not use a causal mask. We pass each point embedding through a shared linear layer to get two predictions per point: one for the probability of the point being a salient point $\hat{p}$ and the other for the offset $\hat{\phi}_i = (x_i, y_i, z_i)$ between the point position and target waypoint position, illustrated by the middle "Prediction" panel of Fig. 3.

Instead of using a hard one-hot target for salient point prediction, we construct a soft salient map over points where the probability of each point is given by:

$$p_i \propto \|c_i - c_k\|_2 \text{ if } \|c_i - c_k\|_2 \leq r \text{ else } 0 \tag{1}$$

Here, $k$ is the index of the point selected by the user and $r$ is a hyperparameter defining a neighborhood of points that are salient. Within this radius, the probability of saliency decreases with distance

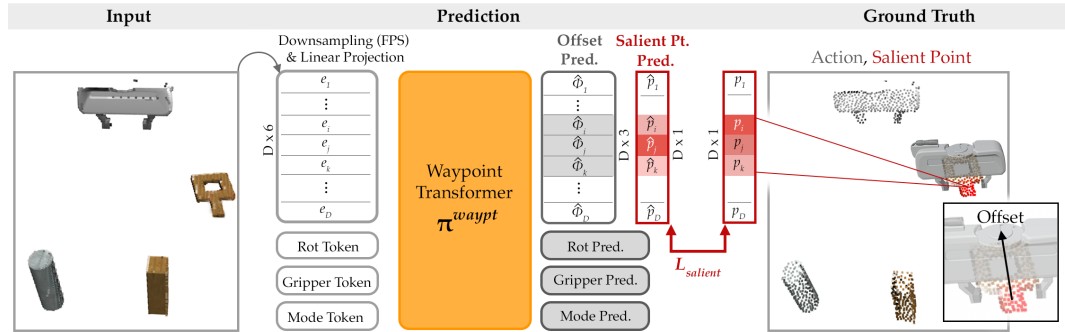

Figure 3: **SPHINX-Waypoint Architecture & Training Objectives:** SPHINX takes downsampled point clouds as input, generating per-point tokens $e_i$, and uses a Transformer-style architecture to predict *salient points* and waypoint actions (position, orientation, gripper state). Specifically, SPHINX predicts the waypoint's positional component as an offset from a salient point. The model outputs a per-point translational offset $\phi_i$, but we only penalize the offset loss on *salient points* (shaded) during training. Salient point prediction is supervised using cross-entropy loss ($L_{\text{salient}}$) between predicted $\hat{p}_i$ and ground truth $p_i$ salient probabilities.

to the ground-truth point. In practice, we set to $r$ to 5cm in all of our experiments. This target distribution of salient points is illustrated by the red points on the right-most panel in Fig. 3, where different shades of red represent the magnitudes of the probability $p_i$. We train saliency prediction using cross entropy loss on the predicted salient probability $\hat{p}_i$ :

$$L_{\text{salient}} = -\sum_i p_i \log \hat{p}_i \tag{2}$$

The goal for the offset prediction is to recover the target end-effector location relative to a predicted salient point, which intuitively aims to ground the waypoint prediction in task-relevant features. This can potentially provide more spatial grounding and awareness that can help with the policy more effectively generalizing. Since we only care about the predicted offset from points with *high* saliency at inference time, we only penalize the offset loss on the points that matter, whose ground-truth salient probabilities $p_i$ are nonzero. Assuming $\boldsymbol{\xi}$ is the position of the target end-effector location, $\boldsymbol{\varphi}_i$ is the location of the $i$-th point, the loss for offset prediction $\hat{\phi}_i$ is:

$$L_{\text{offset}} = \frac{1}{\sum_i \mathbb{1}\{p_i > 0\}} \sum_i \mathbb{1}\{p_i > 0\} \cdot \|\boldsymbol{\xi} - \boldsymbol{\varphi}_i - \hat{\phi}_i\|^2 \tag{3}$$

The prediction targets for rotation, gripper state and next mode are the same regardless of which point is selected. Therefore, we predict them from the output embedding of their respective tokens. We use the mean squared error (MSE) loss for rotation on euler angles $L_{\text{rot}}$, binary cross entropy (BCE) loss for the binary gripper state $L_{\text{gripper}}$, and negative log-likelihood (NLL) loss for next mode classification $L_{\text{mode}}$. Since these losses are standard, we move them to Appendix A for conciseness. Finally, the full waypoint loss is the sum of all terms, and we find that a simple unweighted sum works well in practice, eliminating the need of additional hyperparameters:

$$L_{\text{waypoint}} = L_{\text{salient}} + L_{\text{offset}} + L_{\text{rot}} + L_{\text{gripper}} + L_{\text{mode}} \tag{4}$$

By default, the waypoint dataset only consists of $\{(o_t, a_t, \texttt{waypt}, w_t, z_t)\}$ for timesteps $t$ where the demonstrator explicitly specified a waypoint. However, because we use a controller to move the end-effector between its current pose and the waypoint via $k_t$ interpolated actions, we can treat interpolated steps as additional data points to train on with the *same* target waypoint action label. We can expand each observation-action pair to a sequence:

$$\{(o_t, a_t, \texttt{waypt}, w_{t'}, z_{t'}) \mid t' \leq t \leq t' + \alpha k_{t'}\} \tag{5}$$

where $\alpha$ is a hyperparameter specifying how much of the interpolated data to train on.

### 4.3 THE DENSE POLICY OF SPHINX

The waypoint policy $\pi^{\texttt{waypt}}(w_t|o_t)$ guides the robot to reach objects in a desired pose. However, long-horizon tasks often contain sub-tasks like insertion or alignment that require finer-grained actions. These parts of the task would be easier to accomplish through direct, per-time-step teleoperation, rather than a sequence of waypoints. To address this, we train a dense policy,

$\pi^{\texttt{dense}}(a_t|o_t^{\texttt{wrist}})$, which takes over near a salient point to perform precise manipulation before handing control back to the waypoint policy. Note that this policy uses the wrist camera instead of the point cloud $o_t^{\texttt{pcd}}$ for higher-resolution perception of close-up object details. Ignoring global observations also encourages generalization to scene variations since $\pi^{\texttt{dense}}$ only operates locally.

We instantiate the dense policy of **SPHINX** with diffusion policy (Chi et al., 2023) which has been shown to work well in a wide range of manipulation tasks. To allow the dense policy to switch back to waypoint mode, we augment its action with an additional mode prediction dimension that predicts the next mode $m_t \in \{0, 1, 2\}$ corresponding to $\{\texttt{waypt}, \texttt{dense}, \text{or } \texttt{terminate}\}$ modes. Fig. 1 illustrates how the dense policy fits into the entire **SPHINX** framework. We train the dense policy using the entire dataset and use the interpolated steps to augment our data. This provides the policy with more data and encourages it to be robust to slightly early or late mode switches.

## 5 EXPERIMENTS

In this section, we evaluate how **SPHINX**'s attention to salient points and hybrid policy architecture impacts its performance and generalization on a suite of four challenging real-world and two simulated tasks. In all experiments, we assume access to two external camera viewpoints and a wrist-mounted camera on a Franka Panda robot. See Appendix B for implementation details.

### 5.1 EVALUATION ON PRECISE AND LONG-HORIZON TASKS

We first evaluate **SPHINX**'s performance on complex, long-horizon tasks that demand precision. We hypothesize that by interleaving waypoint actions predicted from point clouds, and dense actions predicted from close-up wrist-camera images once near a salient point, **SPHINX** will more effectively be able to complete this class of tasks compared to baselines which do not exploit salient points or a hybrid mode-switching policy. To assess this, we consider 3 challenging real-world tasks which we found to be infeasible to teleoperate in waypoint-mode alone due to the required degree of precision and reactivity: *Cup Stack* (30 demonstrations), *Train Track* (30 demonstrations), and *Coffee Making* (60 demonstrations). See Fig. 5 for visualizations of each task and example rollouts. For each task, we consider a large and diverse space of initial configurations (Fig. 6), varying the relative locations of objects (cups, mugs, coffee pods, machine, train track).

**Baselines:** We compare against three baselines, the first being **Diffusion Policy (DP)** from Chi et al. (2023) with images from all three cameras as input. We intend for this baseline to demonstrate the benefit of waypoint modes (or lack thereof) for challenging manipulation tasks. The second baseline is **HYDRA** (Belkhale et al., 2023), a hybrid IL policy which takes images from all three views as input and outputs waypoint and dense actions. Given that it is image-based, HYDRA uses a *multiheaded* policy with a shared image encoder as input to waypoint, dense, and mode prediction heads. Its waypoint prediction head outputs a waypoint *without* any intermediate salient representation. This baseline aims to explore the effects of separate input modalities (point cloud vs. wrist image) for waypoint and dense modes, as well as the benefit of using salient points for grounding. The original HYDRA implementation used a simple MLP for the dense head, but we update it to a diffusion policy for fairness. The last baseline is **Fine-tuned OpenVLA**, where we fine-tune the recent OpenVLA model (Kim et al., 2024) on our single-task datasets. We use it as an independent baseline given the recent trend of finetuning models trained on large prior datasets. Due to its design restrictions, this model can only take a single third-person image as input.

In Fig. 4 (left), **SPHINX** achieves the best performance across all three challenging tasks. Fine-tuned OpenVLA struggles to achieve the required precision, lacking mode switches and close-up wrist images. HYDRA shows nonzero performance but suffers from inaccurate waypoint predictions without salient point attention and point clouds. Diffusion policy performs the best among the baselines but struggles to generalize across initial configurations without the waypoint mode. As shown in Fig. 6, **SPHINX** generalizes better across various object placements, while baselines tend to memorize a few arrangements. Overall, these tasks are highly unforgiving of grasping failures and imprecision. Baseline methods particularly struggle to make progress in the train and coffee tasks where early mistakes (missed grasps or placements with the mug, coffee pod, train) derail an entire rollout. **SPHINX**'s use of salient point attention and close-up wrist images allows the policy to carefully proceed through difficult task phases, leading to higher success across scenes.

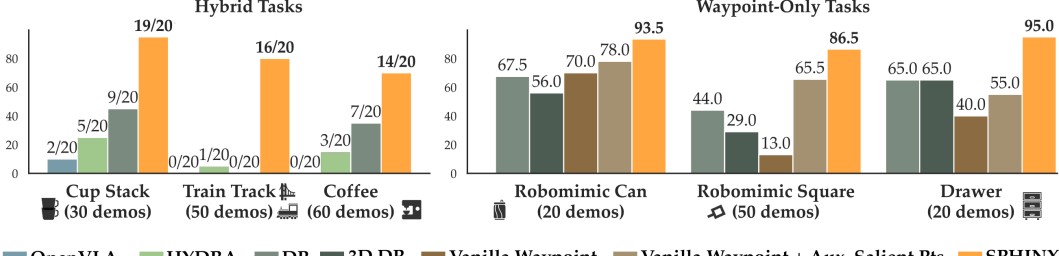

Figure 4: **Success Rates Across Tasks:** Left: SPHINX outperforms image-only dense baselines (OpenVLA, diffusion policy) as well as a hybrid baseline (HYDRA) across 3 challenging real-world tasks (Fig. 5) collected with *hybrid* mode teleoperation. *Train Track* requires a degree of precision that baselines lack, while SPHINX's use of salient points and hybrid actions enables precise, long-horizon manipulation. Right: SPHINX performs 1.6× better than the SoTA image or point-cloud based diffusion policies across tasks teleoperated in only *waypoint* mode. Comparisons with the two vanilla waypoint baselines also show that both saliency prediction and the relative waypoint action representation contribute to SPHINX's strong performance.

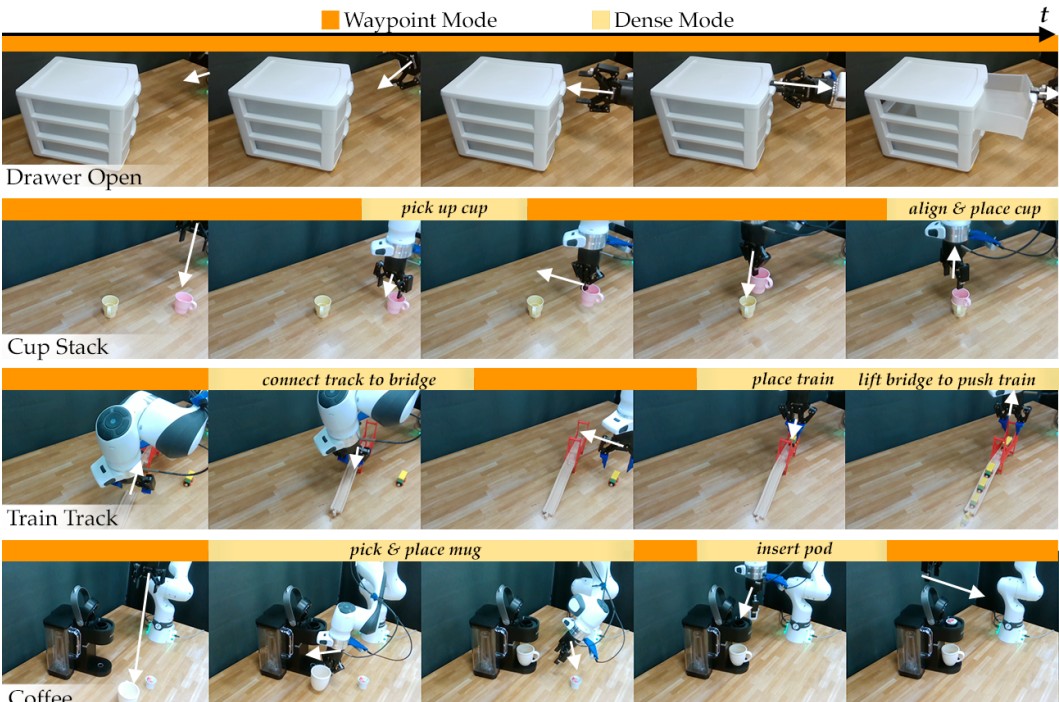

Figure 5: SPHINX **Rollouts**: We evaluate SPHINX across a suite of challenging real-world tasks subject to wide initial state variations. SPHINX's waypoint mode alone is precise enough to handle tasks like drawer opening, while the full hybrid policy leverages different action modes to tackle complex tasks such as cup stacking, building and playing with a toy train set, and making coffee.

## 5.2 WAYPOINT POLICY ABLATIONS

Although SPHINX's performance in challenging real-world tasks relies on the dense policy, the waypoint policy is crucial for its strong performance and generalization as it reliably guides the robot to task-relevant locations. In this section, we validate the design choices behind SPHINX's waypoint policy through ablations and comparisons against state-of-the-art (SoTA) IL policies. To isolate the impact of the waypoint policy from that of the dense policy, we conduct these experiments exclusively on tasks that can be teleoperated solely in *waypoint mode*, without the use of dense mode.

We posit that SPHINX can achieve a higher task success rate across a wide range of initial spatial configurations by effectively using waypoints (via salient points and offsets) to reduce the action prediction horizon and maintain precision and spatial awareness. We consider three tasks, one real

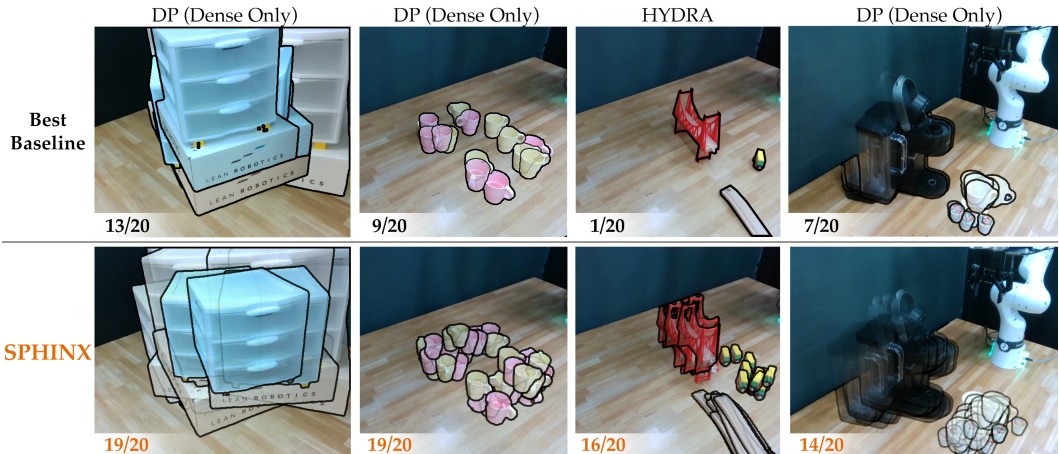

Figure 6: **Distribution of Successful Rollouts**: Across 4 real-world tasks, we visualize the initial state distribution for successful trials across methods by overlaying segmented initial images. Notably, **SPHINX** handles the widest degree of spatial variety, while achieving a much higher task success rate compared to the most competitive baseline. For particularly difficult tasks like *Train Track* and *Coffee*, the baselines tend to memorize motions for particular object arrangements (i.e. mug behind pods) rather than generalize to spatial variations.

| Method | Drawer Unseen height | Drawer Novel viewpoint | Cup Stack Visual distraction | Cup Stack Novel viewpoint |
|---|---|---|---|---|
| Diffusion Policy | 4/10 | 1/10 | 1/10 | 0/10 |
| **SPHINX** | **9/10** | **9/10** | **8/10** | **9/10** |

Table 1: **OOD Results**: We compare **SPHINX** to DP on *Drawer* and *Cup Stack* across various unseen scenarios. In *Drawer*, **SPHINX** successfully opens the drawer at unseen heights by attending to the handle as a salient point, while DP struggles. In *Cup Stack*, **SPHINX** maintains strong performance despite visual distractors, as its salient points focus on the cups, and the wrist-only dense policy ignores surrounding scene changes. Finally, **SPHINX** generalizes to unseen camera viewpoints in both tasks, whereas DP's image-based approach suffers.

task of opening an articulated drawer (*Drawer*, 20 demonstrations) and two simulated environments in Robomimic (Mandlekar et al., 2021) (*Can*, 20 demonstrations, and *Square*, 50 demonstrations).

**Baselines:** We compare **SPHINX** against four baselines. The first two are SoTA IL policies in robotics — dense-only image-based **Diffusion Policy** (Chi et al., 2023) and its point-cloud based extension **3D Diffusion Policy (3D DP)** (Ze et al., 2024). We train them on the interpolated data $\{o_t, a_t\}$. In addition to RGB images from the same external cameras used by the point cloud-based methods, the image-based DP also incorporates wrist camera inputs to ensure optimal performance.

Additionally, we consider two waypoint baselines that can also be seen as separatet ablations over the importance of salient point and offset prediction. **Vanilla Waypoint** uses the same input and Transformer backbone as **SPHINX** but it removes both salient point and offset prediction. It instead adds a Translation Token, similar to the Rotation Token, to the Transformer and directly predicts the target translation from the output of that token using MSE loss. **Vanilla Waypoint + Auxiliary Salient Points** predicts the translation the same way as vanilla waypoint but adds the salient point prediction of **SPHINX** as an auxiliary task. This is essentially **SPHINX** without offset prediction.

**Results:** In Fig. 4 (right), we find that **SPHINX** achieves $1.6\times$ better performance than the best SOTA image-based or 3D-based baseline policy. The vanilla waypoint baseline is only slightly favorable in the simplest *Can* task, but notably worse than the diffusion policies on the more complex *Square* and *Drawer* tasks. This is interesting as it suggests that the shorter prediction horizon of waypoint policies alone is not necessarily a benefit without predicting salient points. Adding the salient point prediction as an auxiliary task improves the performance of vanilla waypoint without changing the action representation, especially in *Square* which requires attending to highly precise object parts such as a small tool handle. This suggests the utility of salient points for encouraging better action prediction. **SPHINX**'s dominant performance over the two waypoint baselines suggests the effectiveness of anchoring waypoint action prediction *relative* to *salient points*.

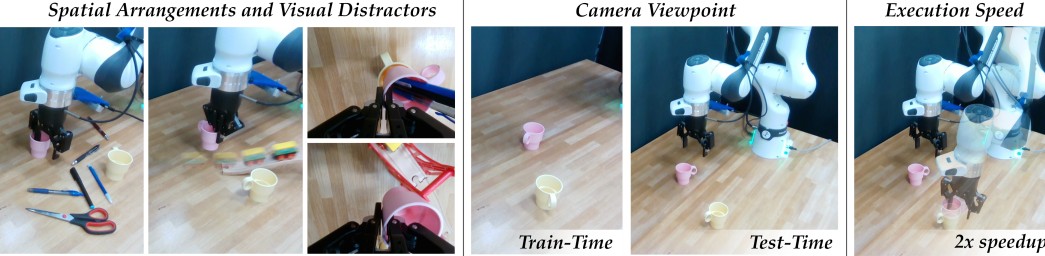

Figure 7: **Generalization Capabilities**: SPHINX generalizes across multiple axes on *Cup Stack*, handling static (e.g., office supplies) and dynamic (e.g., moving toy train) distractors (see website). Using calibrated point clouds as input further allows for generalization to unseen camera viewpoints at test time, and the way-point controller enables *accelerated* execution speeds, completing *Cup Stack* in 9.2 s. on average.

### 5.3 VISUAL GENERALIZATION

We next evaluate SPHINX's ability to handle *visual* rather than only spatial generalization on *Drawer* and *Cup Stack*. As seen in Table 1, SPHINX demonstrates a promising degree of generalization to *visual distractors* during execution, and *unseen third-person camera viewpoints* (Fig. 7), retaining high performance. This is likely enabled by using viewpoint-agnostic point clouds (assuming calibration), salient points encouraging the policy to ignore distractors, and wrist-camera images in dense mode which are unaffected by surrounding changes in the surrounding scene. Diffusion policy heavily overfits to the training scenes and suffers a noticeable performance drop.

### 5.4 EXECUTION SPEED

A key advantage of SPHINX over dense-only methods is its decoupled waypoint and dense policies. While dense methods are tied to the speed of actions recorded during data collection, SPHINX uses a waypoint controller that allows flexible execution speeds at test time. We specifically collect all data across all tasks using a controller limited to a maximum delta of 1 cm at 10Hz. After training SPHINX and dense-only diffusion policy on *Cup Stack*, we perform 10 trials of the task, where we compare SPHINX implemented with a $2\times$ sped-up controller (2 cm maximum delta at test-time) to DP trained on the $1\times$ data. SPHINX completes the task in an average of **9.2 seconds**, a $1.7\times$ speedup over diffusion policy (15.6 seconds). While further speed increases led to controller imprecision, SPHINX has potential for even faster execution on more capable hardware.

## 6 LIMITATIONS AND CONCLUSION

SPHINX demonstrates strong performance and generalization across a range of tasks, but our policy is not without failures. The majority of SPHINX's failures stem from the dense policy being slightly imprecise for grasping or manipulation. Although we mitigate this by using the dense policy only for short horizons near salient points, performing the "last mile" of precise insertion or alignment remains challenging for some tasks. Additionally, our data collection interface uses a linear controller to reach waypoints. This currently limits SPHINX to fairly quasistatic tasks without fast, dynamic movements. Finally, the performance of our waypoint policy is limited by the quality of the input point cloud. We currently perform a one-time calibration procedure to obtain multi-view extrinsics and point clouds, but sensor noise and calibration error is not completely avoidable.

To summarize, we present SPHINX, a visuomotor IL policy which learns to perform complex manipulation tasks from a limited amount of demonstrations while generalizing across many axes: novel spatial arrangements, visual distractors, novel viewpoints, and even customizable execution speeds (Fig. 7). SPHINX achieves this using a hybrid policy architecture that takes point clouds and wrist-images as input, and outputs waypoints and dense actions guided by salient points. SPHINX achieves an average success rate of **86.6%** across 2 simulated and 4 real-world high-precision, long-horizon tasks, including making coffee and assembling a train set with several pieces. Our policy outperforms state-of-the-art IL baselines by **41.1%** on average across **440** real world robot trials. Avenues for future work include improving dense mode manipulation with additional sensing modalities such as tactile sensors, extending our data collection interface to support dynamic manipulation tasks, and deploying SPHINX in the wild across a range of real-world environments.

## ACKNOWLEDGMENTS

This work is in part supported by funds from NSF Awards 2132847 and 2218760, the Office of Naval Research under ONR N00014-22-1-2293, DARPA TIAMAT, DARPA #W911NF2210214, as well as Intrinsic and the Stanford Institute for Human-Centered Artificial Intelligence (HAI). Priya Sundaresan is supported by an NSF GRFP. We would like to thank Suvir Mirchandani, Suneel Belkhale and Yuchen Cui for their helpful feedback and suggestions.

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

# A  ADDITIONAL LOSSES FOR THE WAYPOINT POLICY IN SPHINX

In Section 4.2 we define the salient point prediction loss and offset loss for SPHINX. Here we complete the full loss definitions for the waypoint policy. Recall that the waypoint policy needs to predict translation $\hat{\boldsymbol{\xi}}$, rotation $\hat{\boldsymbol{\alpha}}$, binary gripper state $\hat{g}$ and next mode $\hat{m}$. As described in the main paper, the translation prediction is decomposed to first predict salient points (Eq. (2)) and then predict offset $\hat{\boldsymbol{\phi}}$ w.r.t. the salient point (Eq. (3)). We represent rotations in Euler angles and loss is mean squared error (MSE) between the prediction $\hat{\boldsymbol{\alpha}}$ and target $\boldsymbol{\alpha}$:

$$L_{\text{rot}} = \|\boldsymbol{\alpha} - \hat{\boldsymbol{\alpha}}\|^2. \tag{6}$$

Although MSE on Euler angle ignores the wrap-around effect, i.e. $-\pi$ and $\pi$ represent the same rotation but the loss is not 0, we choose it for its simplicity and find it to work well in practice. Other representations for rotation and their corresponding losses, such as quaternion, should work similarly. The gripper state is binary with 1 for open and 0 for close. Assuming $g$ is the ground-truth gripper state and $\hat{g}$ is the predicted probability of the gripper being open, then the gripper loss is a binary cross entropy loss:

$$L_{\text{gripper}} = g \log \hat{g} + (1 - g) \log(1 - \hat{g}). \tag{7}$$

The waypoint policy also need to predict the next mode, which is a three-way classification among candidates $\{\texttt{waypt}, \texttt{dense}, \texttt{terminate}\}$. We train it via negative log-likelihood:

$$L_{\text{mode}} = -\log \hat{m} \tag{8}$$

where $\hat{m}$ is the predicted probability for ground-truth mode $m$. Finally, the full waypoint loss is the sum of all terms, and we find that a simple unweighted sum works well in practice, eliminating the need of additional hyperparameters:

$$L_{\text{waypoint}} = L_{\text{salient}} + L_{\text{offset}} + L_{\text{rot}} + L_{\text{gripper}} + L_{\text{mode}} \tag{9}$$

# B  IMPLEMENTATION DETAILS OF SPHINX

## B.1  WAYPOINT POLICY

The waypoint policy in SPHINX is using a Transformer to predict salient probability and offset per point, as well as to predict rotation, gripper and mode using additional tokens similar to the [CLS] token in visual classification. The Transformer has 6 layers and each layer has 512 embedding dimensions over 8 attention heads. We remove positional embeddings from Transformer as the point cloud input has no ordering. We set dropout to 0.1 for all Transformer blocks to avoid overfitting. We optimize the waypoint policy with Adam (Kingma & Ba, 2015) optimizer with base learning rate $1e{-}4$ and cosine learning decay over the entire training process, i.e. decaying to 0 at the end of training. We clip the gradient with maximum norm 1. We set batch size to 64. We also maintain an exponential moving average (EMA) of the policy with the decay rate annealing from 0 to 0.9999. We use the final EMA policy in all evaluations without any further model selection. All waypoint policies are trained for 2000 epochs.

As mentioned in the Section 4.2, we use observations from interpolated steps as data augmentation for waypoint training, a technique we refer to as temporal augmentation:

$$\{(o_t, a_t, \texttt{waypt}, w_{t'}, z_{t'}) \mid t' \leq t \leq t' + \alpha k_{t'}\} \tag{10}$$

where $t'$ is the initial step when the demonstrator specified the current waypoint, $k_{t'}$ is the total number of steps it takes for the controller $\mathcal{C}$ to execute this waypoint. Essentially we train $\pi^{\texttt{waypt}}(w'_t | o_t)$ on the initial $o_{t'}$ as well as the intermediate observations $o_{t'+1:t'+\alpha k_{t'}}$. Here $\alpha = 0$ means no temporal augmentation and $\alpha = 1$ means training on the entire waypoint segment. In practice, we find that setting $\alpha = 0.2$ strikes a balance between sufficient augmentation while avoiding interpolated observations that occur too late in a waypoint segment, which can cause the policy to confuse the current target waypoint with the next one. As a concrete example, the waypoint dataset for the Robomimic Square task contains 50 demonstrations, each containing 6 waypoints. The raw dataset for training the waypoint policy contains 300 examples. With the temporal augmentation, the dataset now contains roughly 1800 examples, increasing the amount of data by 6 times. All the waypoint

| Env | Can (20 demos) | | | Square (50 demos) | | |
| --- | --- | --- | --- | --- | --- | --- |
| Waypoint Method | Vanilla | Vanilla (+ Aux SP) | **SPHINX** | Vanilla | Vanilla (+ Aux SP) | **SPHINX** |
| w/o Temp. Aug. | 68% | 70% | 77.5% | 8% | **68%** | 75.5% |
| w/ Temp. Aug. | **70%** | **78%** | **93.5%** | **13%** | 65.5% | **86.5%** |

Table 2: Waypoint Policy Ablations

policies, including vanilla waypoint baselines in Section 5.2 are trained with temporal augmentation $\alpha = 0.2$ as it improves the performance for all of them. The performance of each waypoint policy with and without temporal augmentation is listed in Table 2. We can see that temporal augmentation improves the performance of waypoint policies in five out of six scenarios and achieves similar performance in the remaining one. **SPHINX** benefits the most from the augmentation technique, with 13.5% average improvement across the two environments.

Apart from the temporal augmentation, we also apply random translation augmentation to the point cloud input and target action. The amount of random translation is sampled uniformly from $[-5\text{cm}, 5\text{cm}]$ along $x, y, z$-axes. We follow the common practice to crop the point cloud to remove points outside of the workspace but do not apply any other vision pipelines (i.e. no object detection nor segmentation) to preprocess the point cloud.

### B.2 DENSE POLICY

The dense policy in **SPHINX** is a diffusion policy. We closely follow the original implementation of Chi et al. (2023). Specifically, we use ResNet-18 (He et al., 2016) encoder to process the wrist image and append the proprioceptional to the image embedding before feeding it to a 1-D convolutional UNet for action denoising. The diffusion policy is trained with DDPM to predict the noise given the noisy action as input and observation as context. We follow the best practices of training it using Adam (Kingma & Ba, 2015) with weight decay, cosine learning rate schedule and take the exponential moving average of the policy as final policy for evaluation. Our implementation is able to reproduce the results on Robomimic from the original paper.

## C EXTENDED RELATED WORK

**Relationship between Salient Point and Affordance**. Affordance learning is a common concept in robotics manipulation. It can refer to classifying the nature of interaction for certain object points (e.g., a tool handle is "graspable," a button is "pressable") (Borja-Diaz et al., 2022), or more broad action candidates or success possibilities associated with objects (Ahn et al., 2022). Salient points in **SPHINX** can be thought of as a form of per-point affordance, but must be combined with offsets to specify how the end-effector should interact with a given point. Critically, we demonstrate that the per-point affordances in **SPHINX** allow the policy to focus on task-relevant features and avoid paying attention to arbitrary objects, leading to robust execution in the presence of visual distractors.

**Action Representation in Robotics.** Several works have also considered using classification instead of regression for action prediction in the imitation learning setting. PerAct (Shridhar et al., 2022) divides the entire 3D workspace into voxels and converts the end-effector pose prediction problem into classification over the fixed set of voxels. The precision of prediction depends on the granularity of the voxel, and the number of action candidates grows cubically in the number of voxels, making it challenging for tasks that require high-precision. Act3D (Gervet et al., 2023) performs coarse-to-fine scoring for "ghost points", which addresses the granularity issue of PerAct, but still lacks any kind of explicit intermediate representation such as salient points. In comparison, **SPHINX** first predicts the salient point, a point that *physically exists* in the input point cloud, through classification and then predicts an offset w.r.t. the salient point to recover the full action. This allows **SPHINX** to be arbitrarily precise without incurring the high cost of having fine-grained voxels.

SGRv2 (Zhang et al., 2024) is a recent work that uses per-point offset prediction for robotic manipulation. It predicts actions as per-point offsets for *all* points, and uses a weighted average to get the final action output. In contrast, **SPHINX** utilizes *salient points* learned from human labels as the anchor for offset prediction. Due to the existence of salient points, **SPHINX** applies the offset loss only to the proximal points of the "demonstrator-specified salient point", i.e. the red points in Fig. 3 and Eq. (3). This an easier task for the neural network since it does not need to allocate capacity to predict offsets for points far from the salient points. Additionally, the salient point learning ob-

jective is one of the main strengths of **SPHINX** . The ablation in the right panel of Fig. 4 shows that the classification loss is helpful. Even having it as an auxiliary task significantly improves the performance of the "vanilla waypoint" policy. On the implementation side, **SPHINX** uses a GPT-2 style Transformer while SGRv2 uses PointNeXt as the backbone. Although both methods should work well with either architecture choice but we think it is interesting to see a generic architecture not specialized for point clouds to perform quite well in these tasks.

# D IMPLEMENTATION AND PERFORMANCE OF BASELINES

In this section we discuss the steps we have taken to ensure that the performance of our baselines is valid and discuss potential reasons for the low performance for baselines like Hydra and OpenVLA.

**Diffusion policy (DP):** Our DP implementation is able to reproduce the reported results from the original paper and it achieves the reported performance on Robomimic Can and Square with 200 demonstrations from the original Robomimic dataset. It uses the same set of cameras (3rd person and wrist) as **SPHINX**. In the **SPHINX** experiment we collected 50 demonstrations for the Square and we have verified that the DP trained on the Sphinx dataset ($44\%$, 50 demonstrations) is similar to the one trained on 50 demonstrations from the Robomimic dataset ($45\%$, 50 demonstrations).

**3D diffusion policy (DP3):** Our implementation closely follows the one from the original codebase. In the original paper, DP3 is not evaluated on the Robomimic benchmark. We find it to perform similarly as DP on the real world Drawer task but perform slightly worse than DP on the Robomimic benchmarks and other more complicated real world tasks. This is reasonable given that DP3 purely conditions point clouds constructed from 3rd person cameras and does not use wrist camera images. The close-up information from the wrist camera images is crucial for Robomimic tasks as well as our real world tasks. In comparison, **SPHINX** switches to a wrist-view image based diffusion policy for fine-manipulation, resolving the issue of a lack-of-detail in point cloud inputs.

**Hydra:** We modernize Hydra by using the diffusion policy as its dense policy while keeping the rest of the implementation as close to the original design as possible. The low performance of Hydra seems unreasonable at first glance, but it can be explained via a close look at their original results and our ablations. Our coffee-making task is similar to the one in Hydra. However, the original Hydra paper collected 100 demonstrations with little variation on the location of the coffee machine, and put the cup and coffee pod on a shelf to make them easier to pick up. In our case, we randomize the initial location and orientation of the cup, pod and coffee machine, and only use 60 demonstrations. Therefore, it is reasonable to expect much lower performance for Hydra on this task. Hydra's waypoint branch is similar to our "vanilla waypoint", i.e. directly predicting a target pose via regression, but uses images instead of point clouds. From the ablation (Fig. 4, right panel), we see that this vanilla waypoint policy is noticeably worse than plain diffusion policy on the hard task (Square) that requires precision. Therefore, considering that Hydra's waypoint policy is worse than diffusion policy, it is not surprising to see that the full Hydra policy (vanilla waypoint + diffusion policy) performs worse than diffusion policy in the real world tasks.

**OpenVLA:** We follow the instructions and code provided by the OpenVLA authors and apply everything as in their fine-tuning script (which does use image augmentation). We believe the poor performance of OpenVLA is not surprising for a few reasons. From Figure 5 (left half) of the OpenVLA paper, we see that fine-tuned OpenVLA is *worse* than Diffusion Policy in 2 out of the 3 cases in single-task setting on a Franka Panda robot arm. In general, OpenVLA's distribution of pre-training data relies heavily on the Bridge Dataset, which has a major embodiment gap compared to our Franka dataset. OpenVLA is only compatible with a single 3rd person view, and cannot leverage a close-up wrist image view. On all of the real world tasks we did, the wrist camera view is crucial to sensing end-effector object alignment for precise manipulation. For example, for cup stack (the easiest of the hybrid tasks), even being off by a centimeter can lead to knocking over one or both of the cups. Thus, OpenVLA's poor performance is unsurprising given its limited observation space. Other recent work (Black et al.) reports similar trends on OpenVLA where a fine-tuned OpenVLA performs poorly, achieving $0\%$ success rate on 4 out of 5 settings and $\approx 35\%$ on 1 setting.

