# OpenReview forum: "What's the Move? Hybrid Imitation Learning via Salient Points"
_ICLR.cc/2025/Conference — ICLR 2025 Poster_

### Official Review · Reviewer_gz3J · 2024-10-20

**Soundness:** 4
**Presentation:** 4
**Contribution:** 4
**Rating:** 8
**Confidence:** 5

**Summary:**

This paper presents SPHINX, a novel framework for imitation learning in robotics. SPHINX integrates a hybrid approach by leveraging multimodal inputs (point clouds and wrist camera images) and using both waypoint-based and dense action predictions. It introduces a method for switching between these modalities and action types depending on the task phase, which enables better spatial and visual generalization. The paper demonstrates how SPHINX can handle long-horizon tasks with high precision, achieving superior results compared to state-of-the-art baselines, with an average performance improvement of 41.1%.

**Strengths:**

1. Effective Multimodal Input Usage: The use of point clouds for spatial awareness in long-range movements and wrist-camera images for precision is well-conceived. By switching between input modalities at the appropriate phases of a task, the system exploits the strengths of each modality to improve both efficiency and accuracy.

2. Empirical Performance: The paper presents strong experimental evidence, showing significant performance improvements across multiple real-world and simulated tasks. Specifically, it demonstrates SPHINX’s ability to generalize across novel configurations, camera viewpoints, and visual distractors, which is critical for practical deployments in environments.

**Weaknesses:**

1. Similarity to SGRv2: I noticed that the waypoint policy in this paper is quite similar to the one proposed in SGRv2 [1]. However, the authors do not discuss this in the related work section, nor do they compare their method against SGRv2 in the experiments. Including a detailed comparison or discussion would help clarify how SPHINX improves upon or differs from existing approaches in waypoint-based policies.

[1] Zhang, Tong, et al. "Leveraging Locality to Boost Sample Efficiency in Robotic Manipulation." arXiv preprint arXiv:2406.10615 (2024).

---

2. Insufficient Visual Generalization Experiments: The experiments on visual generalization are somewhat limited. It remains unclear whether SPHINX’s salient point-based design allows the policy to generalize to different objects within the same category (e.g., cups of varying colors or shapes). Additionally, it would be valuable to test SPHINX’s robustness to changes in environmental factors such as background, lighting, or table textures, which are common in real-world settings. Expanding the generalization tests to cover these variations would provide a more comprehensive evaluation of SPHINX’s capabilities.

3. Limited Exploration of Dynamic Tasks: While SPHINX excels at quasistatic tasks, its performance on dynamic tasks (e.g., tasks requiring fast, reactive control) is not explored. The reliance on a linear controller for waypoints may limit the system’s applicability in environments requiring fast, non-linear movements.

**Questions:**

For questions and concerns, please refer to the weaknesses section above. Overall, I believe this is a strong paper that effectively leverages the advantages of both waypoint and dense policies. The experimental results significantly outperform the next best state-of-the-art IL baseline, and the writing is clear, making the paper easy to follow and understand.

---

> ### Author Response · Authors · 2024-11-24
> **Rebuttal**
>
> Thanks for your insight review and positive feedback! We have incorporated discussion on SGRv2 in our updated paper (Appendix C) following your suggestion.
>
> > **Similarity to SGRv2: I noticed that the waypoint policy in this paper is quite similar to the one proposed in SGRv2 ...**
>
> Thanks for pointing out missing related work. SGRv2 is a very recent work from CoRL2024 and we were unaware of it at the time of writing. We can definitely extend related work to cover it. Here we briefly list out the differences between SGRv2 and our work, as well as some thoughts on how the two works would compare in terms of performance:
> * SGRv2 operates on a mixture of point cloud and image data. It predicts actions as per-point offset for _all_ points, and uses a weighted average to get the final action output.
> * Although both works represent the actions as offsets w.r.t. points, Sphinx utilizes “salient point” (learned from human labels) as the anchor of the action, and casts part of the action prediction, a typical regression only problem, to **classification + regression** to make it more precise and easy-to-learn. The ablation in the right panel of Figure 4 shows that the classification loss is helpful, that even just having it as an auxiliary task significantly improves performance of the “vanilla waypoint”.
> * Due to the existence of salient points, Sphinx applies the offset loss only to the proximal points of the “demonstrator-specified salient point”, i.e. the red points in Figure 3 and Eq (3). This makes it easier for the neural network to learn since it does not need to allocate capacity to predict offset for points far from the salient points. It also allows the network to have different position targets for different parts of the points, e.g. the offsets near the top drawer correspond to actions for opening the top drawer while the offsets near the bottom drawer correspond to actions for opening the bottom drawer. In early development, we found applying the offset loss to all points noticeably hurt the performance as it lowers the precision for the most critical points that are used during inference.
> * On a minor note, Sphinx uses a GPT-2 style Transformer while SGRv2 uses PointNeXt as backbone. This is an implementation detail and both methods should work well with either architecture choices but we think it is interesting to see a generic architecture not designed for point cloud to perform quite well in these tasks.
>
> We notice that SGRv2 also evaluates on the Square task from the MimicGen, which is the same as the one in Robomimic. With 50 demonstrations, SGRv2 only achieves a 2.8% success rate, which is significantly lower than Sphinx (86.5%) and the popular diffusion policy (44%) baseline. We have trained the diffusion policy using 50 demonstrations from the original Robomimic dataset and it achieves comparable performance (45%) as the one trained on our data, indicating that the data quality is similar. Therefore, from the published result it seems that there would be a big gap between SGRv2 and Sphinx on Square.
>
>
> > **Insufficient Visual Generalization Experiments**
>
> Although we demonstrate SPHINX across perturbed viewpoints and in the presence of distractor objects, we agree that we have yet to explore more drastic visual variations. One promising avenue for future work is leveraging recent foundation models like MOLMO [1] which have demonstrated impressive performance for zero-shot semantic keypoint detection (i.e. “point to the ___” in an image, as shown here). Instead of directly predicting salient points and offsets jointly, one possible extension is to first infer salient points via models like MOLMO, and condition the waypoint actions on these. We could still collect data for and train SPHINX largely in the same way. However, this would open up SPHINX’s visual generalization capabilities at test-time, since these foundation models could generalize better for salient point detection on completely unseen object instances, with which we can condition the policy.
>
> [1] Deitke, Matt, et al. "Molmo and pixmo: Open weights and open data for state-of-the-art multimodal models." arXiv preprint arXiv:2409.17146 (2024).
>
>
> > **Limited Exploration of Dynamic Tasks**
>
> We acknowledge that in our current work SPHINX uses a linear controller for waypoint reaching, limiting its capacity for dynamic tasks. To relax this, future work could predict spline-based trajectories to reach a waypoint or replace SPHINX’s linear controller with a learned [1] or engineered motion planner for dynamic, collision-aware reaching. SPHINX’s dense policy can in theory be able to handle dynamic motions as long as the hardware permits, so the truly dynamic parts of a task (i.e. wiping, sweeping, higher-velocity actions) can also be learned by the dense policy in the future.

---

> > ### Comment · Reviewer_gz3J · 2024-11-27
> >
> > My main question was about the comparison between the waypoint policy and SGRv2. The author responded comprehensively, and I have no further questions.

---

### Official Review · Reviewer_57XM · 2024-11-04

**Soundness:** 2
**Presentation:** 3
**Contribution:** 2
**Rating:** 6
**Confidence:** 3

**Summary:**

The paper introduces SPHINX which interleaves 3D saliency-based waypoint prediction with fine-grained policy that uses wrist camera image. The proposed method, SPHINX, is compared against SOTA imitation learning methods for robotic manipulation like Diffusion Policy, 3D Diffusion Policy, HYDRA, and also a fine-tuned openVLA. The authors also compare SPHINX with the ablated waypoint models without saliency points. The experiments were done on 4 real world tasks (cup stack, train track, coffee, open drawer) and two simulated tasks in Robomimic environemnt.

**Strengths:**

The proposed method seamlessly integrates multiple concepts in robotic manipulation such as waypoints, affordances, and coarse-to-fine, 2D+3D hybrid approach. These techniques were often explored separately in prior works. Leveraging these techniques, SPHINX effectively solves long-horizon tasks (due to the waypoint approach) and is robust to OoD settings (due to the 3D + saliency structure). Empirical results are strong enough to support these claims. SPHINX show substantial improvements over previous SOTA methods. Overall, the paper is of high quality.

**Weaknesses:**

The primary weakness of this paper lies in its lack of novelty. While the paper successfully integrates many concepts into one, each concept is not novel. For instance, guiding fine-grained manipulation policies with coarse waypoints is not a new approach. Similarly, using saliency prediction to assist manipulation has been widely addressed in prior works, often categorized as affordance-based approach. It is well known that 3D-based policies are inherently more robust than 2D models in OoD scenarios such as existence of distracting objects.

Additionally, there are several keypose-based methods (which are similar to waypoints) that use 3D point clouds, including PerAct, Act3D, 3D Diffuser-Actor, RVT-1/2. It is unclear why the waypoints of SPHINX is better than that of these models.

Lastly, the saliency-based waypoint approach has clear limitations. While it is indeed a very useful inductive bias for many tasks, it may harm the performance in certain cases, particularly when the grippers are far away from any point cloud in the scene. This isn't true for more general methods like Diffusion Policy. Thus, it’s unsurprising that SPHINX outperforms Diffusion Policy, for its added structure and inductive bias. Furthermore, SPHINX relies on additional human-annotated keypoints. While collecting such data is not particularly difficult, this limits SPHINX being applicable to tasks in which the concept of keypoint doesn't really make sense or is unidentifiable for human.

Overall, while this paper presents a well-executed combination of existing ideas, it appears rather incremental. The experimental results are promising, making it a valuable contribution to a robotics-focused conference. However, the contribution may not be sufficient to recommend acceptance at an ML conference, where significant methodological novelty is typically expected.

**Questions:**

1. Is there any positive transfer between the dense policy part and the waypoint prediction part? Or are these just separate models with no synergy in terms of learning?

2. What specific advantage does SPHINX’s waypoint approach offer? I would expect SOTA keypose models like PerAct, Act3D, 3D Diffuser-Actor, and RVT-1/2 to outperform the proposed waypoint transformer. These models are more structurally sophisticated and battle-tested. They also incorporate locality bias, which I believe would provide benefits similar to those of the paper's saliency-based approach. Adding displacement to saliency essentially assigns affordance to specific, localized areas. Furthermore, these models do not require human-annotated saliency, and outputs SE(3) keypose which would be much stronger than R^3 waypoints.

---

> ### Author Response · Authors · 2024-11-24
> **Response to Reviewer 57XM**
>
> ### Re. Novelty:
>
> > The primary weakness of this paper lies in its lack of novelty. Guiding fine-grained manipulation policies with coarse waypoints is not a new approach. Similarly, using saliency prediction to assist manipulation has been widely addressed in prior works, often categorized as affordance-based approach.
>
> We appreciate your feedback, though we don’t agree on the argument for lack of novelty. Our novelty primarily lies in these two aspects:
>
> **Waypoint Action Decomposition**
> 1) We decompose waypoint action prediction (often a global regression task which can be difficult to learn precisely for far-away targets) into a **salient point classification** (softmax over points in the point cloud) and **offset relative to the salient point** (an easier, more local regression task. In comparison:
> - PerAct[1] omits offsets and instead casts action prediction as a classification problem over voxels, and the precision of prediction depends on the granularity of the voxel grid. Increasing this granularity not only significantly increases the training cost, but also makes the prediction task much harder as the number of action candidates grows cubically in the number of voxels.
> - Act3D[2] performs coarse-to-fine scoring for “ghost points”, which addresses the granularity issue of PerAct, but still lacks any kind of explicit intermediate representation such as salient points.
> - Affordance learning can refer to classifying the nature of interaction for certain object points (e.g., a tool handle is "graspable,” a button is "pressable") [3], or more broad action candidates or success possibilities associated with objects [4]. **Salient points in SPHINX can be thought of as a form of per-point affordance,** but must be combined with offsets to specify how the end-effector should interact with a given point. Critically, we demonstrate that the per-point affordances we learn in SPHINX allow the policy to focus on task-relevant features and avoid paying attention to arbitrary objects, leading to **robust execution in the presence of visual distractors.**
>
> **Hybrid Input AND Output Representations**
>
> 2. To our knowledge, **SPHINX is the first IL agent to be able to interchange both its input/output representations on the fly depending on the task phase, which is crucial for success in precise, long-horizon tasks.**
> - Prior works such as Hydra have proposed using a hybrid action space of waypoint/dense actions for IL, but only consider a fixed input space of images. SPHINX performs **5.4** times better than Hydra on complex real-world tasks, demonstrating the benefits of using point cloud observations in combination with our novel waypoint action representation
> - Several prior works propose waypoint-only policies for manipulation with 3D representations as input; although such representations provide rich spatial information, they do not capture object details at a sufficient resolution necessary for accomplishing any one of the real world tasks we consider in this paper. For example, in the train track task (better viewed through the video from website https://sphinx-il.github.io/), the policy needs to identify the “jigsaw puzzle”-like structure on the bridge and track to properly connect them. Point clouds from consumer-grade depth cameras do not provide a sufficiently high resolution to capture such information and there is also the occlusion issue that prevents the 3rd person camera from seeing the gripper-object interaction clearly. Any policy solely based on point clouds will struggle with tasks that require close-up observation, motivating SPHINX’s dense policy.
>
> Given these clarifications, we respectfully ask the reviewer to reevaluate our novelty contribution and kindly point out missing prior works if either one of the contributions mentioned above has been well-studied previously. We also plan to add these discussions to the paper to better emphasize our contribution.
>
> [1] Shridhar, Mohit, Lucas Manuelli, and Dieter Fox. "Perceiver-actor: A multi-task transformer for robotic manipulation." Conference on Robot Learning. PMLR, 2023.
>
> [2] Gervet, Theophile, et al. "Act3d: Infinite resolution action detection transformer for robotic manipulation." arXiv preprint arXiv:2306.17817 (2023).
>
> [3] Borja-Diaz, Jessica, et al. "Affordance learning from play for sample-efficient policy learning." 2022 International Conference on Robotics and Automation (ICRA). IEEE, 2022.
>
> [4] Ahn, Michael, et al. "Do as i can, not as i say: Grounding language in robotic affordances." arXiv preprint arXiv:2204.01691 (2022).

---

> > ### Author Response · Authors · 2024-11-24
> > **Response to Reviewer 57XM (cont.)**
> >
> > ### Re. OOD Generalization
> >
> > > It is well known that 3D-based policies are inherently more robust than 2D models in OoD scenarios such as the existence of distracting objects.
> >
> > We completely agree that 3D-based policies inherently provide richer spatial information that can promote better spatial generalization. Several prior works, however, have shown visual distractors can significantly cause any visuomotor policy to degrade in performance, whether the policy is image-based [5] or point-cloud based [6,7]. For instance, [6] trains a point-cloud and language-conditioned IL agent for tabletop manipulation which drops in performance from 80% to 47.5% success across 8 real-world tasks, when faced with unseen clutter. Similarly, a point-cloud based BC agent in [7] drops in performance by 14% subject to distractors for a range of kitchen tasks. **We posit that SPHINX’s visual generalization capabilities are thus not only enabled by point clouds, but crucially by also enforcing attention to salient points during training.** This encourages the policy to learn to recognize task-relevant features, even in the presence of distractors.
> >
> > [5] Burns, Kaylee, et al. "What Makes Pre-Trained Visual Representations Successful for Robust Manipulation?." arXiv preprint arXiv:2312.12444 (2023).
> >
> > [6] Parashar, Priyam, et al. "SLAP: Spatial-Language Attention Policies." 7th Annual Conference on Robot Learning. 2023.
> >
> > [7] Torne, Marcel, et al. "Reconciling reality through simulation: A real-to-sim-to-real approach for robust manipulation." CoRL 2024.
> >
> > ---
> >
> > ### Re. Comparison to Keypose-Based Methods:
> >
> > > Additionally, there are several keypose-based methods (which are similar to waypoints) that use 3D point clouds, including PerAct, Act3D, 3D Diffuser-Actor, RVT-1/2. It is unclear why the waypoints of SPHINX is better than that of these models.
> >
> > Thank you for pointing out these additional methods; we have since added discussion of them to an expanded related works section in Appendix C (changes highlighted in blue).
> >
> > We have added a comparison with PerAct, also requested by Reviewer 4Uf5, and we report the following results on the RoboMimic Can/Square tasks.
> >
> > | Method                  | RoboMimic Can | RoboMimic Square |
> > |-------------------------|---------------|------------------|
> > | PerAct                 | 0.77          | 0.12             |
> > | Vanilla Waypoint       | 0.70          | 0.13             |
> > | Vanilla Waypoint - Aux SP | 0.78       | 0.655            |
> > | **SPHINX**                 | **0.935**         | **0.865**            |
> >
> > For the simpler Can task, PerAct achieves a success rate of 77%, comparable to the Vanilla Waypoint baseline. However, the Square task, which requires higher precision for grasping the tool and aligning it carefully over the pole, presents a greater challenge. Here, **PerAct achieves only a 12% success rate, likely due to the voxelized input space introducing imprecision from discretization.** Even the Vanilla Waypoint approach, which uses point clouds and directly predicts waypoints, achieves a similar success rate of 13%. Incorporating salient point classification as an auxiliary task significantly improves performance to 65.5%. This large performance gap suggests the benefit of salient point prediction as an auxiliary task, which PerAct, Act3D, 3D Diffuser-Actor, and RVT 1/2 critically lack. Finally, SPHINX, which leverages both salient point detection and offset prediction, achieves a success rate of 86.5%, demonstrating the effectiveness of our waypoint parameterization in tasks requiring a high degree of precision.
> >
> > We also note that while we empirically found our waypoint policy architecture to work well for a range of high-precision tasks, SPHINX is agnostic to the *exact* choice of waypoint policy implementation. In the future, we can take advantage of the most performant waypoint architectures and adapt them to handle salient points.
> >
> > ---
> >
> > ### Questions
> >
> > > Is there any positive transfer between the dense policy part and the waypoint prediction part? Or are these just separate models with no synergy in terms of learning?
> >
> > The waypoint and dense policies are separate models, each designed for distinct input modalities—point clouds for waypoint predictions and wrist camera images for dense actions. These input modalities inherently require different architectures, so we treat the policies as separate. **Although the policies operate independently during inference, their synergy lies in the overall framework.** Each policy addresses specific aspects of the task: waypoint policies handle long range spatial movements, while dense policies manage precise, local manipulation. Each policy also predicts the next mode of execution, enabling seamless transitions that make progress by leveraging the strengths of both at the right moments.

---

> > > ### Author Response · Authors · 2024-11-24
> > > **Response to Reviewer 57XM (cont.)**
> > >
> > > ### Questions (cont.)
> > >
> > > > What specific advantage does SPHINX’s waypoint approach offer? I would expect SOTA keypose models like PerAct, Act3D, 3D Diffuser-Actor, and RVT-1/2 to outperform the proposed waypoint transformer.
> > >
> > > PerAct, Act3D, 3DDA, and RVT-1/2 directly output waypoint actions with no intermediate representation, and 3D Diffuser-Actor exploits locality bias, but still does not leverage any explicit representation like salient points. Instead, SPHINX breaks the waypoint prediction task into two steps: salient point classification and offset regression. This design takes advantage of the following:
> > >
> > > 1. Neural networks are often **more effective at classification than regression**,
> > > 2. **Salient points encourage the policy to recognize task-relevant features of the point cloud,** even in the presence of **visual distractors.**
> > > 3. **Salient point classification as an auxiliary task improves waypoint prediction quality, leading to significantly higher task success rates;** this is demonstrated by the substantially higher performance of the Vanilla Waypoint Aux. SP approach over the Vanilla Waypoint baseline, which omits salient points and offsets, in RoboMimic.
> > >
> > > We have since clarified the distinction between these approaches and SPHINX in Appendix C (Extended Related Work).
> > >
> > > ---
> > >
> > > *How can SPHINX be adapted to avoid human-annotated saliency?*
> > >
> > > Although our data collection interface makes salient points easy to specify on the fly (see the “SPHINX Data Collection Interface” section on website https://sphinx-il.github.io/), we agree that avoiding manual annotation would be preferable. **In the future, we are excited to leverage foundation models to potentially automate salient point annotation.** For example, recent models such as MOLMO [8] demonstrate strong performance in zero-shot semantic keypoint detection (i.e. “point to the ___” in an image, as shown here). This could be used to infer salient points automatically, and only have the demonstrator specify the corresponding waypoint action (offset + orientation + gripper state).
> > >
> > > [8] Deitke, Matt, et al. "Molmo and pixmo: Open weights and open data for state-of-the-art multimodal models." arXiv preprint arXiv:2409.17146 (2024).
> > >
> > > ---
> > >
> > > > Furthermore, these models (PerAct, Act3D, 3D Diffuser-Actor, and RVT-1/2) do not require human-annotated saliency, and outputs SE(3) keypose which would be much stronger than R^3 waypoints.
> > >
> > > We would like to clarify that **SPHINX does output full 6DoF end-effector poses** just as these methods do, but we simply parameterize the positional component of waypoint actions via salient points and offsets. Apologies for the confusion – though we used the term “waypoint” for consistency with Hydra, we can instead refer to these as “keyframes,” etc. in a camera-ready version to convey they capture the 6D pose.
> > >
> > > ---
> > >
> > > > These models are more structurally sophisticated and battle-tested
> > >
> > > We’d like to better understand which aspects of these works you are referring to as more structurally sophisticated and battle-tested. If the latter refers to rigor of evaluation, we make a substantial effort to go beyond the quasistatic tabletop settings explored in PerAct, Act3D, 3DDA, and RVT-1/2 and demonstrate SPHINX’s impressive real-world performance on challenging tasks that demand higher precision, reactivity, and longer-horizon reasoning, such as coffee-making and train track assembly (best viewed [here](https://sphinx-il.github.io/)).

---

> ### Comment · Reviewer_57XM · 2024-11-26
>
> I appreciate your clarifications and the additional experiments. I have raised my scores as the following major concerns have been addressed:
>
> 1. **Benefit of hybrid approach over existing keypose-based approaches.**
> The authors conducted several additional experiments including a comparison with PerAct and an ablation study on a vanilla waypoint model without saliency prediction. These experiments demonstrate the benefits of the saliency-based approach. Furthermore, the authors added an extended related works section in Appendix C. This clarifies the motivation behind SPHINX over existing methods. Lastly, the authors made a fair argument that classification is easier than regression, postulating that SPHINX's generalization capability might not be solely attributable to the 3D representation but also to saliency modeling.
>
> 2. **Lack of Novelty.** I agree with the authors that switching between fine-grained image-based policy and coarse-grained keypose/waypoint models in a real-time closed loop is underexplored. I had underestimated this novelty. It provides a more natural solution to precision tasks than the multiresolution modeling found in recent keypose-based approaches.
>
> 3. **Limitation of saliency-based waypoint approach.**
> This concern was mainly due to my misunderstanding that the waypoints were 3D points rather than 6-DoF (or 6+action_dim) poses. The authors have promised to address this potential confusion by using the term "keyframes" instead of "waypoints":
> > ... though we used the term “waypoint” for consistency with Hydra, we can instead refer to these as “keyframes,” etc. in a camera-ready version to convey they capture the 6D pose.
> >
> I would, however, suggest using "7-DoF waypoint" or "(6+1)-DoF waypoint," as "keyframe" could also be confused as a video frame. That said, the decision is entirely up to the authors.
>
>
> **[Remaining Concerns]:** I am still not fully convinced of the necessity of explicit saliency modeling. The authors claim that explicitly modeling saliency in the (6+1)-DoF waypoint prediction provides three benefits over existing keypose prediction/generation methods:
>
> 1. Classification is easier than regression.
> 2. Saliency points encourage recognition of task-relevant local points, improving robustness to visual distractors.
> 3. Saliency prediction serves as an auxiliary learning objective.
>
> While I agree with these claims, I am still not sure whether explicit saliency modeling is truly necessary. The authors benchmarked PerAct and attributed its low success rate to its low spatial resolution. However, this has already been addressed in more recent works like RVT-2, Act3D, and 3D Diffuser-Actor, which leverages multiresolution approaches and/or image-3D hybrid modeling. It remains unclear whether saliency modeling would still provide significant benefits over these more advanced models in which spatial precision is not a major concern.
>
> In addition, unlike PerAct, these recent models are more sensitive to locality. This can also "encourage the policy to recognize task-relevant features of the point cloud,", achieving similar effects to what explicit saliency modeling aims to accomplish but in a much flexible way. Given that these SOTA keypose models are far more competitive than PerAct, there remains a strong possibility that they could outperform the simple waypoint transformer in SPHINX, even without explicit saliency modeling.
>
> **[Conclusion]:**
> I am more inclined to accept the paper after the rebuttal, as several important concerns have been addressed by the authors. Accordingly, I am revising my recommendation. That said, the concept of switching between waypoint policy and dense policy seems orthogonal to saliency modeling, and I am uncertain about the necessity of the latter when more advanced keypose methods are used for waypoint inference. The authors conducted additional experiments with PerAct, but PerAct is significantly behind recent SOTA methods.

---

### Official Review · Reviewer_B6a8 · 2024-11-04

**Soundness:** 3
**Presentation:** 3
**Contribution:** 3
**Rating:** 6
**Confidence:** 4

**Summary:**

This paper builds on the HYDRA approach by splitting motion into two modes: waypoint-reaching and dense manipulation. The key improvements are: (1) using point clouds instead of RGB images for more robust waypoint localization and using separate PointNet, (2) upgrading the dense manipulation module from a simple MLP to a diffusion policy for smoother control, and (3) introducing "salient points" that explicitly signal when to switch between modes.

The method is evaluated on three challenging, long-horizon real-world tasks, with results showing clear performance gains over baseline approaches. The paper is clearly written, and the experiments are thoughtful, including interesting tests on out-of-distribution scenarios.

**Strengths:**

1. **Enhanced Localization with Point Clouds**: The use of point clouds bolsters waypoint localization, offering a more resilient approach than RGB data alone.
2. **Improved Precision in Dense Manipulation**: Replacing the MLP with a diffusion policy for dense manipulation leads to greater accuracy during complex manipulation phases.
3. **Streamlined Mode Switching**: By introducing salient points, the authors provide a clear and novel mechanism for mode switching, simplifying task segmentation and improving execution flow.
4. **Comprehensive Real-World Tests**: The real-world experiments are particularly strong, presenting complex tasks that highlight the robustness and versatility of the approach.

**Weaknesses:**

1. **Reliance on Salient Points**: The approach depends on expert-determined salient points, which may limit scalability, especially if multiple annotators are involved, as this could introduce inconsistencies in training data.
2. **Lacks Details and Analysis in Experiemnts**: The baseline are too low to tell whether the baselines are correctly run and under fair comparison. It will be better to have some simple tasks which baselines can perform not that bad or provide a failure analysis about how they fail.

**Questions:**

1. **Real-World Experiment Details**: Could the authors share more on how they measure success rates? Do you count broken down points as OpenVLA, or does it only reflect the final task outcome? A failure analysis would also be insightful.
2. **Point Cloud Format**: What specific format of point cloud data does the waypoint transformer use? Are both RGB and XYZ coordinates involved, or only XYZ? As I am curious how the salient point prediction can distinguish the pink and yellow cup.
3. **Handling Distant Salient Points**: When a salient point isn’t near a physical point—like opening an oven or a door—how does the method handle this? If there’s a substantial distance to any nearby object, does the model has to use the mode of dense manipulation?
4. **OpenVLA Training Clarifications**: The performance of OpenVLA seems unexpectedly low in some tasks, like cup stacking. Could the authors clarify their training setup for OpenVLA? For example, was image augmentation applied, and was it trained on a single or multiple tasks?
5. **HYDRA Performance Insights**: It would be interesting to understand why HYDRA, despite using a similar waypoint and dense manipulation modes, performs way less effectively. Does it struggle more in the waypoint phase or the dense manipulation phase?
6. **Camera Viewpoint Calibration In Novel Viewpoint**: Were novel camera viewpoints recalibrated during inference? Additionally, are the training and inference data in the same coordinate system, like the robot’s base?
7. **Waypoint Transformer**: It’s interesting that the waypoint transformer model can effectively handle unordered points with relatively small data. Could you provide more details on the network structure and training? What's more, were any data augmentations applied?

---

> ### Author Response · Authors · 2024-11-24
> **Rebuttal**
>
> Thanks for your insightful feedback! We are happy to add more details and clarify some misunderstandings. We are trying to address all of your concerns below but changed the orders of some questions to make the explanation more coherent.
>
> > **Handling Distant Salient Points: When a salient point isn’t near a physical point—like opening an oven or a door—how does the method handle this? If there’s a substantial distance to any nearby object, does the model has to use the mode of dense manipulation?**
>
> It seems that there is a misunderstanding here which we will address in the revised version through a clearer explanation of salient points versus waypoints. In our design, the salient point **must** be a physical point that exists in the point cloud. For example, when opening an oven/door, the demonstrator clicks on the handle to specify the salient point. The eventual effector pose can be anywhere, like 3 cm or 10 cm away from the salient point along a certain direction. We record the _waypoint action_ as a combination of salient points (a point in the point cloud that physically exists) and offset w.r.t. that point.
> It is a core part of Sphinx that the salient points are physical points in the point cloud and that is what allows us to cast part of the action prediction problem into classification over a set of points. Classification is generally more robust as it encourages competition among the candidates, and it also makes residual offset prediction simpler. It is one of the main insights that motivates the Sphinx method.
>
> > **Reliance on Salient Points**
>
> It is true that Sphinx relies on the annotated salient point for training. But the salient point annotation is organically integrated into the data collection process (better illustrated through videos in the “SPHINX Data Collection Interface” section on the website https://sphinx-il.github.io/). With our GUI, it is not necessarily an overhead to add the salient point. Instead, it may simplify the data collection process compared to the very common “dense” teleoperation interfaces like a spacemouse or VR controller that require constant attention to control the robot precisely. For example, opening a drawer is just 3 mouse clicks, each followed by some click-and-drag to adjust the offset and angle, which is considerably simpler than commanding the robot using a VR controller.

---

> > ### Author Response · Authors · 2024-11-24
> > **Rebuttal cont.**
> >
> > > **Lacks Details and Analysis in Experiments:  The baseline are too low to tell whether the baselines are correctly run and under fair comparison.**
> >
> > We apologize if this wasn’t clear in our paper.  We have added more discussion on the performance of the baselines in the Appendix D. Here we provide a brief comment on the performance of each baseline used in the paper.
> >
> > **Diffusion policy (DP)**: Our DP implementation is able to reproduce the reported results from the original paper and achieves 97% success rate on Can and 92% on Square with **200** demonstrations from the original Robomimic dataset. It uses the same set of cameras (3rd person and wrist) as Sphinx. In the Sphinx experiment we collected 50 demonstrations for the Square and we have verified that the DP trained on the Sphinx dataset (44%, 50 demonstrations) is similar to the one trained on 50 demonstrations from the Robomimic dataset (45%, 50 demonstrations). So we are confident that this baseline works as expected.
> >
> > **3D diffusion policy**: Our implementation closely follows the one from the original codebase. In their original paper, 3D diffusion policy was not evaluated on the Robomimic benchmark. We find it to perform similarly as DP on the real world Drawer task but perform slightly worse than DP on Robomimic benchmarks and other more complicated real world tasks. We believe this is reasonable given that 3D diffusion policy purely conditions point clouds constructed from 3rd person cameras and does not use wrist camera images. The close-up information from the wrist camera images is crucial for Robomimic tasks as well as our real world tasks. In comparison, Sphinx switches to a wrist-view image based diffusion policy for fine-manipulation, resolving the lack-of-detail issue of point cloud inputs.
> >
> > **Hydra**: The low performance of Hydra seems unreasonable at first glance. However, this can be explained via a close look at their original results and our ablations.
> > Our coffee-making task is similar to the one in Hydra. However, the original Hydra paper collected 100 demonstrations with **no** variation on the location of the coffee machine, and put the cup and coffee pod on a shelf to make them easier to pick up. In our case, we randomize the initial location and orientation of the cup, pod and coffee machine, and only use 60 demonstrations. Therefore, it is reasonable to expect a much lower performance for Hydra on this task.
> > Hydra’s waypoint branch is similar to our _vanilla waypoint_, i.e. directly predicting target pose via regression, but uses only images instead of point clouds. From the ablation (Figure 4, right panel), we see that this vanilla waypoint policy is noticeably worse than plain diffusion policy on the hard task (Square) that requires precision. Therefore, considering that Hydra’s waypoint policy is worse than diffusion policy, it is not surprising to see that the full Hydra policy (vanilla waypoint + diffusion policy) performs worse than diffusion policy in the real world tasks.
> >
> > **Re. OpenVLA specifically:**
> > > **The performance of OpenVLA seems unexpectedly low in some tasks, like cup stacking. Could the authors clarify their training setup for OpenVLA? For example, was image augmentation applied, and was it trained on a single or multiple tasks?**
> >
> > **OpenVLA**: We follow the instructions and code provided by the OpenVLA authors and apply everything as in their fine-tuning script (which does use image augmentation). We believe the poor performance of OpenVLA is not surprising for a few reasons.
> > From Figure 5 (left half) of the OpenVLA paper, we see that fine-tuned OpenVLA is **worse** than Diffusion Policy in 2 out of the 3 cases in single-task setting on Franka Panda hardware. In general, OpenVLA’s distribution of pre-training data relies heavily on the Bridge Dataset, which has a major embodiment gap compared to our Franka dataset.
> > OpenVLA is only compatible with a single 3rd person view, and cannot leverage a close-up wrist image view. On all of the real world tasks we did, the wrist camera view is crucial to sensing end-effector object alignment for precise manipulation. For example, for cup stack (the easiest of the hybrid tasks), even being off by a centimeter can lead to knocking over one or both of the cups. Thus, OpenVLA’s poor performance is unsurprising given its limited observation space.
> > Other recent works report similar trends on OpenVLA. For instance, the recently released pi policy [1] (see Figure 7 from https://www.physicalintelligence.company/download/pi0.pdf) with some co-authors from the OpenVLA paper show that a fine-tuned OpenVLA performs poorly, 0% success rate on 4 out of 5 settings and ~35% on 1 setting.
> >
> > [1] pi0: A Vision-Language-Action Flow Model for General Robot Control [Pre-print]

---

> ### Author Response · Authors · 2024-11-24
> **Rebuttal cont.**
>
> > **HYDRA Performance Insights: It would be interesting to understand why HYDRA, despite using a similar waypoint and dense manipulation modes, performs way less effectively. Does it struggle more in the waypoint phase or the dense manipulation phase?**
>
> As discussed above, the waypoint head in Hydra is similar to the vanilla waypoint in ablation, both trained to directly regress a target pose without considering the salient point. Vanilla waypoint struggles in the Square task, suggesting that just doing the waypoint-dense decomposition alone is **not helpful** compared to modern imitation algorithms like diffusion policy. This highlights the value of our novel action representation (salient point classification + offset regression). Hydra struggles a lot in the waypoint phase as its waypoint prediction is much worse than Sphinx. The error of Hydra’s waypoint predictions is often a few centimeters while the error of Sphinx’s waypoint predictions is less than 1cm.
>
> > **Real-World Experiment Details: Could the authors share more on how they measure success rates? Do you count broken down points as OpenVLA, or does it only reflect the final task outcome? A failure analysis would also be insightful.**
>
> We measure success rate as follows:, if the robot successfully finishes all manipulation steps for a task then it is 1, otherwise it is 0. We do not follow OpenVLA’s way of computing scores via partial progress. We have video recordings for all the evaluations, best viewed at https://sphinx-il.github.io/ (you can click on any bar in the “Hybrid Task Results” section to view rollouts for any policy). Different methods fail for all sorts of reasons, failing to pick up the object, cannot navigate to the correct location etc.
>
> > **Point Cloud Format: What specific format of point cloud data does the waypoint transformer use? Are both RGB and XYZ coordinates involved, or only XYZ? As I am curious how the salient point prediction can distinguish the pink and yellow cup.**
>
> As mentioned in the paper, we use colorized point clouds, i.e. each point represented by x,y,z,r,g,b, a tuple of 6 floating points.
>
> > **Camera Viewpoint Calibration In Novel Viewpoint: Were novel camera viewpoints recalibrated during inference? Additionally, are the training and inference data in the same coordinate system, like the robot’s base?**
>
> Yes, the novel camera viewpoints are calibrated. Despite calibration, the novel camera view leads to quite different images (Figure 7, middle panel) as the robot body is mostly out of the view for the training camera while occupying half of the image (and thus accounting for a significant portion of the point cloud input) in the novel inference camera. Yes the coordination system is the same throughout our paper.
>
> > **Waypoint Transformer: It’s interesting that the waypoint transformer model can effectively handle unordered points with relatively small data. Could you provide more details on the network structure and training? What's more, were any data augmentations applied?**
>
> Yes it is indeed interesting that a GPT2 style transformer with no point cloud specific structure can handle points. During early development we also found it worked better than PointNet++. The network structure is similar to GPT2, using the nanoGPT (https://github.com/karpathy/nanoGPT) as reference implementation. As described in Appendix B1, the waypoint transformer has 6 transformer blocks, has no positional embedding for the points, and uses dropout=0.1 for all blocks. We train it with a cosine learning rate schedule and return the exponential running average as the final policy.
>
> As described in the last paragraph of Section 4.2 (right above Section 4.3 on page 6), we use “temporal data augmentation”. At time step t, to reach a waypoint (wt+k), we use a controller that moves the arm over the interval [t, t+1, …, t+k]. Observations at intermediate steps share the same waypoint label (wt+k), allowing us to use these steps for data augmentation. In all experiments, we use the first 20% of interpolated steps (t, …, t + 0.2*k) as additional training data for the waypoint policy. For example, in the Square task, this technique augments 300 demonstrations into a dataset of approximately 1,800 data points. On top of the “temporal data augmentation”, we apply random translations to the input and target, a typical data augmentation technique for point cloud based neural networks. We have since added these clarifications to Appendix B (Implementation Details of SPHINX).

---

> > ### Author Response · Authors · 2024-12-02
> >
> > We would like to thank you again for your constructive feedback. As the review period is approaching an end, we would like to know if there is any outstanding questions that we have not clarified.
> >
> > As a recap, our rebuttal focuses on:
> > 1. Clarifying that the salient point must be a physically existed point in the point cloud, i.e. it handles far-away objects with no problem.
> >
> > 2. We have added additional details on how we ensure our baseline implementations are valid and have commented on why we think the low performance of some baselines are reasonable given either ablations from our paper or evidence from other recent papers. We have video recordings for all the evaluations, best viewed at https://sphinx-il.github.io/ (you can click on any bar in the “Hybrid Task Results” section to view rollouts for any policy).
> >
> > 3. Other experimental details such as no partial scores and using colorized point cloud input.
> >
> > Thanks to your suggestion, these new information would increase the confidence and significance of our experimental results.

---

### Official Review · Reviewer_4Uf5 · 2024-11-05

**Soundness:** 3
**Presentation:** 4
**Contribution:** 2
**Rating:** 5
**Confidence:** 4

**Summary:**

In this work, the authors consider the task of imitation learning in the context of tabletop robot manipulation tasks. Specifically, they investigate the well-known tension between the need for both long-horizon and short-horizon control: e.g. non-dexterous freespace motion for a large fraction of a trajectory, followed by a short dexterous stretch. They articulate the hierarchical nature of many manipulation tasks, and propose to address the hierarchy with a hybrid policy which can switch between two modes: a high-level waypoint prediction which is grounded in a global observation point cloud, and a low-level fine-grained control which operates on wrist camera inputs. They describe a policy architecture and training procedure to fit this paradigm, as well as a data collection/annotation method for generating datasets with required annotations suitable for training. They then demonstrate the efficacy of their policy on 3 custom real-world manipulation tasks.

**Strengths:**

# Originality

* While hierarchical approach has been proposed, for instance in HACMan, NDF, TAX-Pose, etc, and explicit mode-switching between policies has been explored, I haven’t really seen this sort of mode switching in imitation learning for long-ish horizon manipulation.
* Giving the policy a mechanism to change its inputs is a neat design consideration
* First time seeing diffusion policy in a hierarchical context.

# Quality

* The real-world experiments are well-designed, and demonstrate good performance on a relatively small number of demos
* The idea is straightforward and a reasonable solution to this class of quasistatic manipulation problems.
* Decent ablation coverage.
* The design of the visual generalization experiments was interesting.

# Clarity

* Very clear and well-written.

# Significance

* Reasonable factorization of the problem, so might be useful for for practitioners looking to solve this class of problems expediently and don’t have needs for generalizing outside of quasi-static pick-and-place tasks.

**Weaknesses:**

* The model’s hypothesis class (e.g. mode switching between long-distance free space motion, and short horizon fine-grained manipulation) imposes a major constraint on the kinds of problems it can represent effectively. Of course, in the extreme case, either one or the other mode can always be predicted, but in these settings no benefit / insight is offered, and hard attention switching across the inputs limits flexibility.
* The assumptions about dataset preparation are a major weakness/limitation here, for two reasons:
    1. Adding a different label for mode is a major annotation requirement, and can be quite noisy/arbitrary.
    2. The notion of labeled “salient points” may make some sense because humans do attend to salient points reasonably consistently, but generally speaking they are not a consistent/principled thing to expect a human to label - who is to say which specific points should be annotated? Seems quite noisy, and a major source of error.
* Comparisons were somewhat weak.The authors assert that long-horizon goal prediction is not precise enough for this kind of imitation learning, but evidence from e.g. 3DDA, RPDiff, NDF, TAX-Pose, etc. demonstrate that the quasistatic problems considered in the real-world tasks are probably solvable with goal prediction.
* Benchmark tasks are also somewhat weak. While the real-world experiments are good, the authors should compare against other methods on simulated benchmarks (including annotating using their system or a synthetic annotation scheme) such as RLBench where some medium-horizon tasks have been examined with a litany of different approaches.
* There’s not much analysis of sample complexity.
* Several related works are missing / not discussed:
    * Spatial grounded policies: Spatial Action Maps, Transporter Nets, HACMan, 3D Diffuser Actor
    * Relative placement / reasoning: NDF, TAX-Pose, etc.
    * Keyframing: Many of Stephen James’ works on top of RLBench look at long-horizon gripper positional reasoning and keyframing.
    * The “soft salience” mechanism has been used before (e.g. TossingBot, many others), so while it’s pretty standard it would be good to compare / reference
* The variation for initial conditions in the real-world experiments appear somewhat limited, and having the high-level policy run on a global world frame is a bit unfair as a selling point because it doesn’t have to generalize as much by construction (probably could modify the other methods to have similar properties).

**Questions:**

* Can the authors speak to sample complexity analysis for the different methods considered?
* How could this be used / modified to not require human annotation of mode/salient points directly?
* How does this method perform on simulated benchmarks w/ published baselines/metrics?

---

> ### Author Response · Authors · 2024-11-24
> **Response to Reviewer 4Uf5**
>
> We are glad to hear that you appreciate our hybrid policy architecture and the real-world experimental results!
>
> ---
>
> ### Re. flexibility of mode switching
> > Mode switching between long-distance free space motion, and short horizon fine-grained manipulation imposes a major constraint on the kinds of problems it can represent effectively; hard attention switching across the inputs limits flexibility.
>
> We actually intended for our hybrid action space to be able to accommodate a broad range of manipulation tasks that waypoint-only or dense-only policies traditionally struggle with. Waypoint-only policies predict movements too sparsely to handle parts of a task requiring reactive, precise manipulation, like inserting a coffee pod into a slot or assembling puzzle pieces like the train track and bridge together. This limitation also applies to the class of waypoint policies mentioned (3DDA, RPDiff, NDF, TAX-Pose). Conversely, dense-only policies, while flexible, often fail to handle longer-horizon tasks with significant visuospatial variations. For example, in our evaluations, dense policies achieved only 5% and 35% success rates on the sequential train track and coffee-making tasks, respectively, compared to 80% and 70% for SPHINX, demonstrating the effectiveness of hybrid actions. Having access to these two distinct modes actually provides more flexibility, as SPHINX can arbitrarily switch back and forth between dense and waypoint modes at any time.
>
> That said, we acknowledge that in our current work SPHINX uses a linear controller for waypoint reaching, limiting its capacity for dynamic tasks. To relax this, future work could predict spline-based trajectories to reach a waypoint or replace SPHINX’s linear controller with a learned [1] or engineered motion planner for dynamic, collision-aware reaching.
>
> [1] Dalal, Murtaza, et al. "Neural mp: A generalist neural motion planner." arXiv preprint arXiv:2409.05864 (2024).
>
> ---
>
> ### Re. Data Collection Assumptions
>
> > Adding a different label for mode is a major annotation requirement, and can be quite noisy/arbitrary
>
> We would like to clarify that demonstrators do not need to manually annotate mode labels. Modes are automatically determined based on input: dense mode is triggered by joystick movements on the Spacemouse, and waypoint mode by mouse movements in the web UI.
>
> > Generally speaking they are not a consistent/principled thing to expect a human to label - who is to say which specific points should be annotated? Seems quite noisy, and a major source of error.
>
> **Re. noise in salient point annotations:**
> The intent is for the demonstrator to label salient points that are meaningful to the task (e.g., the drawer handle for opening, mug handle for coffee making, etc.). This is often a more clear and direct annotation as opposed to more free-form human annotations such as providing language instructions or narrations. In our case, as our datasets are collected by a single demonstrator, we did not encounter label inconsistency. Additionally, since SPHINX predicts actions as offsets relative to salient points, it can theoretically compensate for somewhat noisy salient points by adjusting the offset predictions accordingly.
>
> That said, we agree that consistent salient point labeling is crucial for scaling up SPHINX data collection. A promising direction for future work is offloading salient point detection to foundation models. For instance, recent models like MOLMO [2] excel at zero-shot semantic keypoint detection (i.e. “point to the ___” in an image, as shown here). This could be used to annotate salient points automatically and only have the demonstrator specify the corresponding waypoint action (offset + orientation + gripper state).
>
> **Re. noise in offset specification:**
> Multimodality in teleoperated data is in general an open challenge works that do crowd-sourcing such as [3-5], regardless of the interface used (VR, puppeteering, Spacemouse), but we believe our data collection interface has several features in place to promote more consistent action labels. Our UI allows the demonstrator to adjust a waypoint action (offset/orientation) as much as they would like via simple clicks and drags on a gripper CAD model in the scene, before sending the controller to reach the final specified waypoint. We see this as a strength of our approach, as it allows the demonstrator to be precise with specification before any waypoint actions are executed (as opposed to retrying after being imprecise). The resulting trajectories are also relatively clean due to the linear controller.

---

> > ### Author Response · Authors · 2024-11-24
> > **Response to Reviewer 4Uf5 (cont.)**
> >
> > [2] Deitke, Matt, et al. "Molmo and pixmo: Open weights and open data for state-of-the-art multimodal models." arXiv preprint arXiv:2409.17146 (2024).
> >
> > [3] Mirchandani, Suvir, et al. "So You Think You Can Scale Up Autonomous Robot Data Collection?." CoRL 2024.
> >
> > [4] Khazatsky, Alexander, et al. "Droid: A large-scale in-the-wild robot manipulation dataset." RSS 2024.
> >
> > [5] pi0: A Vision-Language-Action Flow Model for General Robot Control [Pre-print]
> >
> > ---
> >
> > ### Re. Baselines
> > > Comparisons were somewhat weak.The authors assert that long-horizon goal prediction is not precise enough for this kind of imitation learning, but evidence from e.g. 3DDA, RPDiff, NDF, TAX-Pose, etc. demonstrate that the quasistatic problems considered in the real-world tasks are probably solvable with goal prediction.
> >
> > > The authors should compare against other methods on simulated benchmarks
> >
> > Thank you for the suggestion! We note that NDF and TAX-Pose make 2 important  assumptions which we do not make: (1) only rigid objects are manipulated which allows for the computation of cross-object poses and (2) access to multi-object segmentation. SPHINX makes neither of these assumptions, and specifically considers articulated object tasks such as opening a drawer or closing the coffee machine lid. For this rebuttal, we add a comparison with PerAct which does not make these assumptions (also requested by Reviewer 4), and we report the following results on the waypoint-only RoboMimic Can/Square tasks:
> >
> > | Method                  | RoboMimic Can | RoboMimic Square |
> > |-------------------------|---------------|------------------|
> > | PerAct                 | 0.77          | 0.12             |
> > | Vanilla Waypoint       | 0.70          | 0.13             |
> > | Vanilla Waypoint - Aux SP | 0.78       | 0.655            |
> > | **SPHINX**                 | **0.935**         | **0.865**            |
> >
> > For the simpler Can task, PerAct achieves a success rate of 77%, comparable to the Vanilla Waypoint baseline. However, the Square task, which demands higher precision for grasping the tool and aligning it carefully with the pole, presents a greater challenge. Here, **PerAct achieves only a 12% success rate, likely due to the voxelized input space introducing quantization errors.** This is comparable with the Vanilla Waypoint approach (13%), which uses point clouds and directly predicts waypoints. For the same architecture, incorporating salient point classification as an auxiliary task significantly improves performance to 65.5% (Vanilla Waypoint, Aux SP). This large performance gap suggests the benefit of salient point prediction as an auxiliary task, which 3DDA, RPDiff, NDF, TAX-Pose critically lack. Finally, **SPHINX, which leverages both salient point classification and additionally offset prediction, achieves the highest success rate of 86.5%,** demonstrating the combined effectiveness of both in tasks requiring a high degree of precision.
> >
> > In general, we also note that waypoint-style policies (including 3DDA, RPDiff, NDF, TAX-Pose) by design cannot address tasks requiring precise and reactive movements, such as coffee-making (pod insertion) or train track assembly (placing the train on the bridge). We found that these kinds of fine-grained adjustments are not even reliably teleoperable in waypoint mode alone, due to sensor noise, controller imprecision, and the linear controller limiting reactivity. This highlights the necessity of our hybrid action modes.
> >
> > ---
> >
> > ### Re. Performance & Generalization
> > > There’s not much analysis of sample complexity.
> >
> > While we do not explicitly analyze sample complexity, our empirical results show that, for the same amount of demonstrations, **SPHINX achieves a 53.3% higher success rate than dense-only policies on the hybrid real-world tasks.** Additionally, for a fixed number of demonstrations, SPHINX outperformed Vanilla Waypoint policies that did not leverage salient points and offsets on 3 waypoint-only tasks. This suggests that salient point classification and hybrid action modes promote better sample efficiency.
> >
> > > The variation for initial conditions in the real-world experiments appear somewhat limited
> >
> > We do attempt to vary the initial object poses fairly broadly across the workspace, best visualized at https://sphinx-il.github.io/ (Spatial Generalization Section), though we do acknowledge that the arrangements are more limited for significantly harder tasks like coffee making and train track assembly. We additionally consider a range of visual changes such as novel camera viewpoints, object heights, and distractors. We are excited to further stress-test SPHINX’s generalization capabilities in the future, and are optimistic that using foundation models to detect salient points can allow for more drastic category or instance-level object generalization.

---

> > > ### Author Response · Authors · 2024-11-24
> > > **Response to Reviewer 4Uf5 (cont.)**
> > >
> > > ### Re. Performance & Generalization (cont.)
> > >
> > > > Having the high-level policy run on a global world frame is a bit unfair as a selling point because it doesn’t have to generalize as much by construction (probably could modify the other methods to have similar properties)
> > >
> > > We argue that leveraging point clouds, which naturally encourage better spatial awareness than third person images, is a strength of our approach – a choice that simply is overlooked in many prior works. Attempting to modify them requires the same level of care as in SPHINX, and this choice results in far better generalization and performance across different object spatial arrangements. We do acknowledge and mention that the downside to point clouds is reliance on known camera extrinsics. In the future, it would be interesting to leverage recent works like Dust3r [6] and Monst3r [7], which predict camera extrinsics given RGB images. This would allow us to recover the global world frame for unseen viewpoints at test time, which would retain the generalization benefits of our approach without the overhead of camera calibration.
> > >
> > > [6] Wang, Shuzhe, et al. "Dust3r: Geometric 3d vision made easy." Proceedings of the IEEE/CVF Conference on Computer Vision and Pattern Recognition. 2024.
> > >
> > > [7] Zhang, Junyi, et al. "Monst3r: A simple approach for estimating geometry in the presence of motion." arXiv preprint arXiv:2410.03825 (2024).
> > >
> > > ---
> > >
> > > ### Re. Related Works:
> > > Thank you for putting these additional works on our radar! **We have since included a discussion of Spatial Action Maps, Transporter Nets, HACMan, 3D Diffuser Actor, NDF, TAX-Pose, and 3DDA, RPDiff, and existing waypoint/keyframing policies to Appendix C (Extended Related Work).**

---

> > ### Comment · Reviewer_4Uf5 · 2024-11-27
> >
> > I appreciate the authors' response. However, I am still a bit unconvinced w.r.t. the generalizability of salient point annotation fundamentally from an imitation learning perspective. While i recognize that this decomposition leads to increased performance on the chosen benchmark, I suspect it also would limit larger-scale generalizability. My rationale is this: the human annotator may choose a specific point to anchor to, but there are likely many different salient points in a scene one could anchor on for whatever reason one chooses. If a policy were able to emergently learn this point saliency during interaction, then I'd buy it. But the demonstration space is just too arbitrary - I could choose any point on an object, say that I was anchoring on the rim of the mug instead of the handle of the mug when grasping, in order to come up with a trajectory. What's more, in cluttered scenes, there may be many different possible salient regions affecting the manipulation itself (e.g. grasping a mug under occlusion/infeasible angles/etc). I don't have a problem with annotating saliency per se (a weak/noisy signal in demonstrations is fine) - I simple have a problem that the actual policy itself is GROUNDED at that point (reference frame, coordinates, etc.), and under the IL paradigm this presents a very challenging learning target for generalization under noisy/arbitrary salience points. I would need to see substantially more evidence (of the generalization sort mentioned) to feel comfortable with this sort of demonstration anchoring as a general-purpose solution.

---

### Official Review · Reviewer_7kbH · 2024-11-05

**Soundness:** 3
**Presentation:** 3
**Contribution:** 3
**Rating:** 6
**Confidence:** 5

**Summary:**

This paper introduces an imitation learning framework, SPHINX (Salient Point-Based Hybrid ImitatioN and eXecution), designed to improve the generalization and efficiency of robotic control in complex manipulation tasks. Traditional imitation learning approaches often struggle with diverse visual and spatial variations, especially in tasks requiring long-term planning and precision. SPHINX addresses these challenges by utilizing a hybrid approach that combines two modes of execution: waypoints for large, coarse movements and dense actions for precise, close-up interactions. They show extensive empirical results that demonstrate SPHINX's effectiveness across both real-world and simulated tasks, showing a substantial performance improvement over existing IL baselines.

**Strengths:**

1. This paper proposes a two-stage framework. The first stage leverages salient points for generating long-range movements, specifically as waypoints, followed by interpolation for motion planning and control. The second stage employs an image-based action similar to a diffusion policy, where actions are based on EE poses. This framework is reasonable for multi-stage manipulation tasks that require contact.
2. The paper is relatively complete, covering the proposed method, dataset collection, simulations, and real-world experiments. The baselines are adequately diverse, and the real-world experimental demonstrations are effectively presented.
3. The writing in this paper is clear, with accurate illustrations that aid understanding of the proposed framework. Explanations of the data collection module and experiments are well-supported with corresponding videos.

**Weaknesses:**

1. The novelty of the paper could be further clarified. Overall, applying a hierarchical approach to improve generalization and performance in IL is reasonable. This is novel within IL (only my point, if other reviewers can provide references, I will defer it). However, using a two-stage approach (i.e., long-range navigation + fine-grained manipulation) to handle long-term manipulation tasks is a common strategy, and I suggest the authors add a discussion on this.
2. There are some concerns regarding the visual module. The first stage’s visual module appears to use a transformer for salient point extraction, which resembles affordance learning. It would be beneficial to include a discussion on affordance learning. Additionally, the paper seems to use a limited training set for salient point training, such as 30 samples for cup stacking. Without pre-training, I am skeptical that these datasets can achieve accurate and generalizable visual perception.
3. Concerns related to the robotics aspect. The paper does not seem to mention object segmentation or part-level pose estimation, which also appear to be integrated within the waypoint transformer. It is unclear if this approach is effective or if additional processing was performed in the real-world experiments.

**Questions:**

1. There seems to be a very basic error in the paper: I believe the EE pose should be 6 DoF, but the paper frequently mentions 7 DoF. Is this due to the use of quaternions for rotation, or does it include the EE’s open/close state as an extra DoF?
2. Generalization problem. Does the generalization reported in the paper include generalization across objects of different appearances and shapes, or is it limited to generalization over the initial pose?
3. Mode switching and subtask decomposition. The paper does not seem to address subtask decomposition for long-term tasks, appearing to rely on mode switching. For the two proposed modes, how does the model ensure continuity across temporal sub-steps, and how does it detect the salient point corresponding to the next subtask (distinct from the previous step)?

Overall, this paper is in good shape, with a reasonable method, sufficient experiments, and clear implementation details. If the above concerns and questions can be addressed, I would raise the score.

---

> ### Author Response · Authors · 2024-11-24
> **Rebuttal**
>
> Thank you for your thoughtful review as it truly helps up to improve the paper. Hopefully we can address all of your concerns.
>
> > **The novelty of the paper could be further clarified.**
>
> Our novelty primarily lies in these two aspects:
>
> 1. The novelty of our waypoint policy lies in **decomposing action prediction into salient point classification** (softmax over points in the point cloud) and **offset regression** (predicting a local offset relative to the salient point).
>
> * The salient point classification encourages the policy to focus on task-relevant objects and avoid paying attention to visual distractors.
>
> * Classifying salient points in a point cloud makes the learning objective easier. Classification is generally an easier task for neural networks to learn compared to regression. Additionally, we now only need to regress a small offset within a local neighborhood to recover the positional component of the waypoint, which is easier for objectives like MSE compared to directly trying to regress potentially far away targets. PerAct [1] also casts waypoint prediction as classification, but over voxels which is notably harder; the precision of prediction depends on the granularity of the voxel, and the number of action candidates grows cubically in the number of voxels.
>
> * The benefit of this action decomposition for waypoint policy is clearly demonstrated in ablations (Figure 4, right panel) where we show that SPHINX performs significantly better than the vanilla way of predicting waypoints through regression (vanilla waypoint), even when the baseline has the same amount of supervision signal (vanilla waypoint + aux prediction).
>
> 2. To our knowledge, SPHINX is the first IL agent to be able to interchange **both its input/output representations on the fly** depending on the task phase, which is crucial for success in precise, long-horizon tasks.
>
> * Prior works such as Hydra have proposed using a hybrid action space of waypoint/dense actions for IL, but only consider a fixed input space of images. SPHINX performs **$5.4\times$ better than Hydra** on complex real-world tasks, demonstrating the benefits of using point cloud observations in combination with our novel waypoint action representation
>
> * Several prior works propose waypoint-only policies for manipulation with 3D representations as input; although such representations provide rich spatial information, they do not capture close-up details at a sufficient resolution necessary for accomplishing any one of the real world tasks we consider in this paper. For example, in the train track task (better viewed through the video from website https://sphinx-il.github.io/), the policy needs to identify the “jigsaw puzzle”-like structure on the bridge and track to properly connect them. Point clouds from consumer-grade depth cameras do not provide a sufficiently high resolution to capture such information and there is also the occlusion issue that prevents the 3rd person camera from seeing the gripper-object interaction clearly. Any policy solely based on point clouds will struggle with tasks that require close-up observation, motivating SPHINX’s dense policy.
>
> We also plan to further clarify these two points in the introduction section in a camera-ready version if accepted.
>
> [1] Shridhar, Mohit, Lucas Manuelli, and Dieter Fox. "Perceiver-actor: A multi-task transformer for robotic manipulation." Conference on Robot Learning. PMLR, 2023.
>
>
> > **The first stage’s visual module appears to use a transformer for salient point extraction, which resembles affordance learning. It would be beneficial to include a discussion on affordance learning.**
>
> The connection between "affordances" and "salient points" is a valid question. In robotics, "affordances" can characterize the nature of interaction for certain object points (e.g., a tool handle is "graspable,” a button is "pressable"), or more broader action candidates or success possibilities associated with object points [3]. Salient points in SPHINX can be seen as a form of per-point affordance, but must be combined with offsets to specify how the end-effector should interact with a given point.
>
> For instance, in the coffee-making task, after picking up the mug, the next salient point might lie on the center of the platform of the machine. This provides a meaningful spatial prior over where the policy should interact to place the mug, but does not specify how the robot should orient or position its arm to do so (which a waypoint action specifies). Thus we learn to predict waypoints where the positional component is parameterized as an offset to a salient point.
>
> [2] Borja-Diaz, Jessica, et al. "Affordance learning from play for sample-efficient policy learning." 2022 International Conference on Robotics and Automation (ICRA). IEEE, 2022.
> [3] Ahn, Michael, et al. "Do as i can, not as i say: Grounding language in robotic affordances." CoRL 2022.

---

> > ### Author Response · Authors · 2024-11-24
> > **Rebuttal cont.**
> >
> > > **Additionally, the paper seems to use a limited training set for salient point training, such as 30 samples for cup stacking. Without pre-training, I am skeptical that these datasets can achieve accurate and generalizable visual perception.**
> >
> > Good question. We actually leverage much more than 30 samples via data augmentation techniques, clarified here:
> >
> > 1. **Raw Dataset Format**: Each demonstration consists of multiple waypoint actions, each associated with a particular salient point. For example, to collect a single demonstration for the simulated Square task, the demonstrator first specifies an initial waypoint above the square tool handle, a second waypoint to grasp the tool, a third waypoint to move it above the pole, etc.. Thus, a dataset of 50 Square demonstrations yields ~300 paired data points of the form (observation, waypoint, salient point) with which to train the policy.
> > 2. **Temporal Augmentation** (discussed in Section 4.2, page 6): At time step t, to reach a waypoint (wt+k), we use a controller that moves the arm over the interval [t, t+1, …, t+k]. Observations at intermediate steps share the same waypoint label (wt+k), allowing us to use these steps for data augmentation. In all experiments, we use the first 20% of interpolated steps (t, …, t + 0.2*k) as additional training data for the waypoint policy. For example, in the Square task, this technique augments 300 demonstrations into a dataset of approximately 1,800 data points. The effect of this temporal augmentation is shown in Table 2 of Appendix B.
> > 3. **Positional Augmentation**: We also apply random translations to the point cloud input before feeding it to the network, a common augmentation technique for point clouds. The preprocessing step of FPS (furthest point sampling) can also be seen as a minor form of data augmentation, as the network sees different subsets of the points in different epoch.
> >
> > We have further added the above clarifications to Appendix B.1 (changes highlighted in blue).
> >
> > > **Concerns related to the robotics aspect. The paper does not seem to mention object segmentation or part-level pose estimation, which also appear to be integrated within the waypoint transformer.It is unclear if this approach is effective or if additional processing was performed in the real-world experiments.**
> >
> > There is no object segmentation nor part-level pose estimation in this paper. We simply crop out points that are not part of the workspace of the robot, i.e. only keeping points where: $x \in [-0.5m, 0.5m]$, $y \in [-0.5m, 0.5m]$ and $z$ > 0m
> >
> > > **I believe the EE pose should be 6 DoF, but the paper frequently mentions 7 DoF. Is this due to the use of quaternions for rotation, or does it include the EE’s open/close state as an extra DoF?**
> >
> > We count the gripper dimension action as additional DoF for simplicity.
> >
> > > **Generalization problem. Does the generalization reported in the paper include generalization across objects of different appearances and shapes, or is it limited to generalization over the initial pose?**
> >
> > We assume that the reviewer refers to generalization over the initial pose of the robot end-effector. We did not randomize initial end-effector poses, but we did test generalization over unseen object poses, new camera poses, and visual distractors, all of which are significantly harder since the robot needs to generalize across vastly different spatial arrangements even if starting from the same end-effector pose.
> >
> > We have not tested generalization over different objects yet, as all of the models, including baselines, are not trained to do so, but we fully agree that this is an exciting avenue for future work. In particular, we are excited to potentially offload salient point annotation to LLM/VLMs, which could allow SPHINX to generalize at the object instance or category level.
> >
> > > **Mode switching and subtask decomposition. The paper does not seem to address subtask decomposition for long-term tasks, appearing to rely on mode switching. For the two proposed modes, how does the model ensure continuity across temporal sub-steps, and how does it detect the salient point corresponding to the next subtask (distinct from the previous step)?**
> >
> > During data collection, the demonstrator can seamlessly switch between teleoperating waypoint actions using the web UI or providing dense actions using the Spacemouse. For each timestep, we detect whether movement is sensed on the computer mouse or Spacemouse, allowing us to obtain mode labels for each timestep automatically. From these mode labels, the model learns when to switch modes from the data. Every time the waypoint or the dense policy makes a prediction, it also predicts what the mode should be for the next time step. The model learns that well from data. The dense policy sometimes switches to the waypoint mode a bit early or late, but the waypoint policy is able to generalize over different starting conditions thanks to the temporal data augmentation mentioned earlier.

---

> > > ### Comment · Reviewer_7kbH · 2024-11-28
> > >
> > > Thanks for the explanation. I think most of my concerns are addressed (e.g., visual perception training, generalizability) but some remains tentative. After reading these explanations and other reviews, I will slightly increase the score. Lastly, I encourage the authors to use 6-DoF or (6+1) DoF, as "7-DoF EE" is less-used in robotics.

---

### Author Response · Authors · 2024-11-24
**Shared Response to All Reviewers**

We thank all the reviewers for their valuable feedback and constructive feedback! We are glad to hear that reviewers appreciate SPHINX’s hybrid policy architecture, our impressive performance on challenging real world manipulation tasks, our competitive performance against SoTA baselines, and thorough ablations. We have **updated the paper (changes in blue**) to include more training details on data augmentation (Appendix B), more related work (Appendix C), and discussed the poor performance of some baselines and steps we have taken to ensure that the performance of the baselines is valid (Appendix D).

Below, we highlight shared concerns and our attempts to address them:

---

### Lack of Novelty

We want to clarify that the novelty of our approach is 2-fold:
1. Our **decoupled waypoint action parameterization into salient point classification + offset regression** enables more precise waypoint actions, spatial generalization, and robustness to distractors
2.  **SPHINX is the first IL agent to our knowledge to be able to interchange both its input space and output action space on the fly during execution,** which is critical for progressing through different stages of  long-horizon tasks

### Comparisons Against Alternative Waypoint Architectures
- Reviewers 57XM and 4Uf5 asked about how SPHINX compares against other waypoint policy architectures. Although we have already shown SPHINX’s competitive performance compared to 4 SoTA approaches (Diffusion Policy, 3D Diffusion Policy, Hydra, OpenVLA), we have since added a comparison with PerAct, a SoTA waypoint-only policy which directly outputs waypoint actions (omitting salient points and offsets) given voxelized input. **SPHINX performs 4.2x better than PerAct on the RoboMimic Can and Square tasks.**
- We hypothesize that these performance gains, especially for more precise tasks, are only made possible by SPHINX’s decoupled way of representing waypoint actions as salient points and offsets, which several existing waypoint policy architectures (Act3D, 3DDA, NDF, TAX-Pose) explicitly lack.
- We have since added a discussion around the distinction between these approaches and SPHINX to Appendix C (Extended Related Work).

### Manual Annotation of Salient Points
- Specifying salient points (and corresponding waypoints) in our data collection UI is as simple as a few clicks and drags in **seconds.** We argue that this is both very straightforward to do and worth it for 2 reasons:
 1. The demonstrator no longer has to pay attention to their every single move (as is the case with VR/Spacemouse), since reaching long distances is delegated to a controller and
  2. Salient point annotation is minimal in terms of overhead but serves as a rich source of supervision for the policy.
- Still, we appreciate the reviewer’s suggestions for relaxing the assumption of manual annotation. **In the future, we are excited by potentially offloading salient point detection to recent foundation models such as MOLMO [1]** which excel at zero-shot semantic keypoint detection (i.e. “point to the ___” in an image, as shown here). This could be used during data collection to remove manual salient point annotation, such that the demonstrator only needs to specify the corresponding waypoint action for a predicted salient point, or at test-time to infer salient points on unseen objects and open up SPHINX’s visual generalization capabilities.

[1] Deitke, Matt, et al. "Molmo and pixmo: Open weights and open data for state-of-the-art multimodal models." arXiv preprint arXiv:2409.17146 (2024).

---

In the sections below, we further address each reviewer’s comments and questions point-by-point.

---

### Meta-Review · Area_Chair_LWzH · 2024-12-22

**Metareview:**

The paper introduces SPHINX (Salient Point-Based Hybrid ImitatioN and eXecution), an imitation learning method for manipulation tasks.  It is a hybrid execution model combining coarse waypoint-based navigation and dense action for precise manipulation. It first predicts salient points for long-range movement. Once near a salient point, it switches to predicting dense end-effector movements for precise phases of a task. It is verified empirical validation through real-world and simulated tasks, demonstrating performance improvements over baseline methods.

Strengths:

- The approach it innovative. It provides a clear mechanism for automatic mode switching, potentially addressing limitations other methods face in long-horizon and precision tasks.

- Real-world experiments are extensive and well covered.

 - The writing is clear, and the results are well presented.

(c) Weaknesses:

 - Dependence on expert-determined salient points may limit scalability and introduce inconsistencies across annotators, particularly in multimodal or occluded scenarios.

- While the experiments are compelling, the paper lacks comparison with stronger waypoint-based methods (e.g., RVT-2, 3D Diffuser-Actor).  Especially because one of the major weaknesses pointed out by the authors for the way-point-based method is lack of precision -- "PerAct achieves only a 12% success rate, likely due to the voxelized input space introducing imprecision from discretization." Hence, for this approach to shine in the long run, it must be compared to SOTA waypoint-based, that are supposed to be much more precise.

Final Decision:
Four reviewers tend toward acceptance and one toward rejection. The concerns pointed out by the reviewers are reasonable. However, I think the strengths of the innovative approach outweigh the weaknesses, and so I would recommend acceptance. Also, the paper would benefit greatly from better comparison, as pointed out in Weakness 2, and I hope the authors can add such a comparison in the final version.

**Additional Comments On Reviewer Discussion:**

Overall, this paper was extensively discussed. Key points included:

- Salient Point Annotation (Reviewer 4Uf5, Reviewer B6a8): The authors explained that salient points are automatically set during data collection, reducing annotation noise.

- Baseline Comparisons (Reviewer 4Uf5, Reviewer 57XM): Added PerAct comparisons show SPHINX’s precision advantages. Gaps remain in benchmarking against RVT-2 and 3D Diffuser-Actor.

- Visual Generalization (Reviewer gz3J): The authors plan to use foundation models for better generalization, a promising but future direction.

The paper makes strong contributions to imitation learning. Despite some gaps, the empirical evidence supports its value. I recommend acceptance.

---

### Decision · Program_Chairs · 2025-01-22

Accept (Poster)